# The transcription factors Tfeb and Tfe3 are required for survival and embryonic development of pancreas and liver in zebrafish

Alberto Rissone[1☺], Martina La Spina[1☺], Erica Bresciani[2], Zulfeqhar A. Syed[3], Christian A. Combs[4], Martha Kirby[5], Abdel Elkahloun[6], Vicky Chen[7], Raman Sood[8], Shawn M. Burgess[9], Rosa Puertollano[1*]

1 Cell and Developmental Biology Center, NHLBI, NIH, Bethesda, Maryland, United States of America, 2 Oncogenesis and Development Section, NHGRI, NIH, Bethesda, Maryland, United States of America, 3 Electron Microscopy Core Facility, NHLBI, NIH, Bethesda, Maryland, United States of America, 4 Light Microscopy Core, NHLBI, NIH, Bethesda, Maryland, United States of America, 5 Flow Cytometry Core Facility, NHGRI, NIH, Bethesda, Maryland, United States of America, 6 Microarray and Single-cell Genomics Core, NHGRI, NIH, Bethesda, Maryland, United States of America, 7 Integrated Data Science Services, Research Technologies Branch, NIAID, NIH, Bethesda, Maryland, United States of America, 8 Zebrafish Core, NHGRI, NIH, Bethesda, Maryland, United States of America, 9 Translational and Functional Genomics Branch, NHGRI, NIH, Bethesda, Maryland, United States of America

☺ These authors contributed equally to this work.
* puertolr@mail.nih.gov

## Abstract

The transcription factors TFEB and TFE3 modulate expression of lysosomal, auto-phagic, and metabolic genes to restore energy and cellular homeostasis in response to a variety of stress conditions. Since their role during vertebrate development is less characterized, we used CRISPR/Cas9 to deplete *tfeb*, *tfe3a*, and *tfe3b* in zebraf-ish. The simultaneous lack of these genes compromised embryo survival during early development, with an almost complete lethality of the larvae by 8–10 dpf. The knock-out animals showed apoptosis in brain and retina and alterations in pancreas, liver, and gut. Exocrine pancreas presented the most severe defects, with accumulation of abnormal zymogen granules leading to acinar atrophy in embryos and pancreatitis-like phenotypes in adults; likely due to a block of the autophagy machinery implicated in removal of damaged granules. Knockout animals displayed increased susceptibility to oxidative and heat-shock stress. Our work reveals an essential role of Tfeb and Tfe3 in maintaining cellular and tissue homeostasis during development.

## Author summary

TFEB and TFE3 belong to the MiT/TFE family of basic helix–loop–helix leucine zipper transcription factors and orchestrate cellular responses to a variety of stress conditions, such as nutrient deprivation, oxidative stress, organelle dam-age, and pathogen infection. TFEB and TFE3 translocate from the cytosol to the

**Data availability statement:** All data needed to evaluate the conclusions in the paper are present in the paper and/or the Supplementary Materials. The mass spectrometry proteomics data have been deposited in the ProteomeXchange Consortium via the PRIDE (89) partner repository with the dataset identifier PXD056508. The scRNA-seq data from this publication have been deposited in the NCBI GEO database and assigned the identifier GSE278733. There are no restrictions on data availability.

**Funding:** This research was supported by the National Institutes of Health Intramural Research Program (ZIA HL006075-15 to RP). The funders had no role in study design, data collection and analysis, decision to publish, or preparation of the manuscript. All the authors received a salary from the NIH.

**Competing interests:** The authors declare no competing interests.

nucleus following stress. In the nucleus they regulate expression of hundreds of genes, thus contributing to increase the degradative capability of the cells, restore energy homeostasis, and promote cell survival. The important role played by these transcription factors is evidence by their involvement in several human diseases, including cancer, neurodegeneration, and inflammatory disorders. In this study we assessed the role of TFEB and TFE3 during embryonic development in zebrafish and found that they are essential for tissue homeostasis, adaptation to stress, and ultimately embryo survival.

## Introduction

TFEB and TFE3 are basic helix–loop–helix leucine zipper transcription factors that function as master regulators of cellular adaptation to stress in response to a wide variety of stimuli, including nutrient deprivation, oxidative stress, pathogens, organelle damage, and many others [1]. The control of TFEB and TFE3 intracellular localization is critical to modulate their activity [2]. Rag GTPase-mediated recruitment of TFEB and TFE3 to lysosome membranes under basal conditions, brings them in close proximity to the serine/threonine kinase mTORC1 [3,4]. mTORC1-dependent phosphorylation of TFEB-S211 and TFE3-S321 promotes the binding to 14-3-3, resulting in sequestration of the transcription factors in the cytosol [5–8]. Under stress conditions, inactivation of mTORC1 [5–8], activation of specific phosphatases [9,10], or changes in the conformation of the Rag GTPases [11,12] result in TFEB and TFE3 dephosphorylation and consequent translocation to the nucleus. Additional post-translational modifications, such as acetylation, SUMOylation, oxidation, glutathionylation, polyADP-ribosylation, and itaconatylation, further contribute to regulate their localization, stability, and conformation [13,14]. In the nucleus, TFEB and TFE3 promote expression of genes implicated in multiple pathways, including lysosomal biogenesis, autophagy, metabolism, immunity, cell growth, and differentiation, playing a pivotal role in cellular survival and adaptation to stress, as evidenced by their implication in several human diseases [15,16].

Less characterized is the contribution of TFEB and TFE3 to embryonic development. There is convincing evidence showing that TFEB plays an important role in vascular development [17,18], oligodendrocyte differentiation and myelinization [19], and syncytiotrophoblast formation [20,21]. However, the impossibility to generate whole-animal TFEB/TFE3 double knockout mice, due to the early placentation defects caused by the absence of TFEB [22], has precluded a comprehensive characterization of their function during development. This is an important caveat, given the high level of redundancy displayed by TFEB and TFE3 in different tissues [23–25] and the potential toxicity of the Cre-LoxP system [26]. The compensatory upregulation of other MiT/TFE family members (such as MITF or TFEC) remains also unexplored.

The zebrafish system provides a powerful tool for the analysis of embryonic phenotypes [27]. Compared to mammalian species, zebrafish embryos are very resilient.

They hatch with a yolk sac that is consumed during development and, due to their small size, can exchange oxygen by simple diffusion, surviving until 7 days post-fertilization (dpf), even with major cardiovascular defects [28,29].

In this study, we used the zebrafish system to simultaneously deplete *tfeb* and *tfe3* and found that these transcription factors are essential for larval survival during early development. The knockout animals displayed an ample array of alterations, including increased apoptosis in brain and retina, and obvious morphological alterations in pancreas, liver, and gut. The embryos were also very susceptible to external stressor, such as oxidative stress and heat shock, evidencing the key contribution of *tfeb* and *tfe3* to stress adaptation *in vivo*.

## Results

### Establishment and characterization of zebrafish *tfeb*, *tfe3a,* and *tfe3b* triple knock-out lines

The transcription factors TFEB and TFE3 are considered master regulators of cellular response to stress [30]. To better understand their role during embryonic development, we decided to use zebrafish as animal model to generate knock-outs for these genes. Zebrafish present two copies of the *tfe3* gene *(tfe3a* and *tfe3b)* and a single copy of *tfeb* [1]. In addition, both *tfeb* and *tfe3a* genes present an alternative promoter that gives rise to variants lacking the first exons of the gene [31] (Fig 1A). The short forms lack the N-terminal region of the protein, which includes the Rag-binding and the transactivation domains (Fig 1A). Short forms for the zebrafish *tfe3b* genes have not been described so far [31] and might have been lost through evolution after the last whole genome-duplication event [32].

Initially, we targeted the first exon of the zebrafish *tfeb* gene using a specific gRNA and the CRISPR/Cas9 system and were able to recover a mutated allele carrying a 7 bpDEL (Figs 1A, S1A and S1B); however, the homozygous fish were viable and fertile, and they did not present major phenotypes. Suspecting possible redundancies with other close members of the MiT/TFE family, we decided to use a CRISPR/Cas9 multiplexing approach to simultaneously target multiple exons of *tfeb* and both *tfe3* genes (Fig 1A). For the *tfeb* and *tfe3a* genes, we targeted exons shared by both the long and short isoforms, while for the *tfe3b* gene we coinjected two gRNAs targeting exon 1 and exon 2, respectively (Fig 1A). The gRNAs and CAS9 were directly injected into the original *tfeb* Ex1 mutant line, and we were able to recover multiple heterozygous mutated alleles for each new site and confirm the mutations both at genomic and mRNA level by Sanger sequencing (S1A and S1B Fig). All the confirmed frame-shift mutations were simple deletions (DEL), except for the mutation in the *tfeb* exon 6, which is the result of two distinct events, a 2 bpDEL (GA) and the insertion of 1 base (T) (S1A and S1B Fig). Notably, the *tfe3b* mutations 4 bpDEL and 7 bpDEL (in exon1 and exon 2, respectively) were both on the same allele.

For simplicity, the triple homozygous animals will be henceforth indicated as TKOs. Expression analysis by qPCR showed a strong reduction of all the *tfeb* and *tfe3a* transcripts (long and short) at 2 days post fertilization (dpf) (Fig 1B). Although the *tfe3b* mutations were present at the mRNA level and introduced multiple premature stop codons, the nonsense-mediated mRNA decay (NMD) was not activated (Fig 1B). Since our first attempts to obtain TKO adult fish from intercrosses of triple heterozygous fish, it was evident that the TKO fish presented an extremely reduced size compared to their siblings (Fig 1C). Even after being separated from their siblings following genotyping at three months of development, these very small TKO fish were unable to recover their size, and we almost systematically failed to obtain fertilized eggs in intercross experiments. As shown in Fig 1D, histological analysis of testis of 4-months-old TKO fish did not show any major phenotypes. In contrast, TKO female fish presented a total lack of oocytes in the last phases of maturation, with most of the oocytes blocked in the previtellogenic phases, which explains the failure to get fertilized eggs in intercross experiments. We wondered whether the reduced size and the general limited maturation level of the fish, might be due, at least in part, to food competition [33]. To test this hypothesis, we started to separate the smaller larvae from the rest of the population at ~30 dpf and let them grow in a separate tank. After 3 months of development, we observed an increase in the average size and the number of fertilized eggs in the TKO population isolated from the rest, thus confirming that the food competition was partially responsible for the limited growth and sexual maturation of the larvae. However, whereas

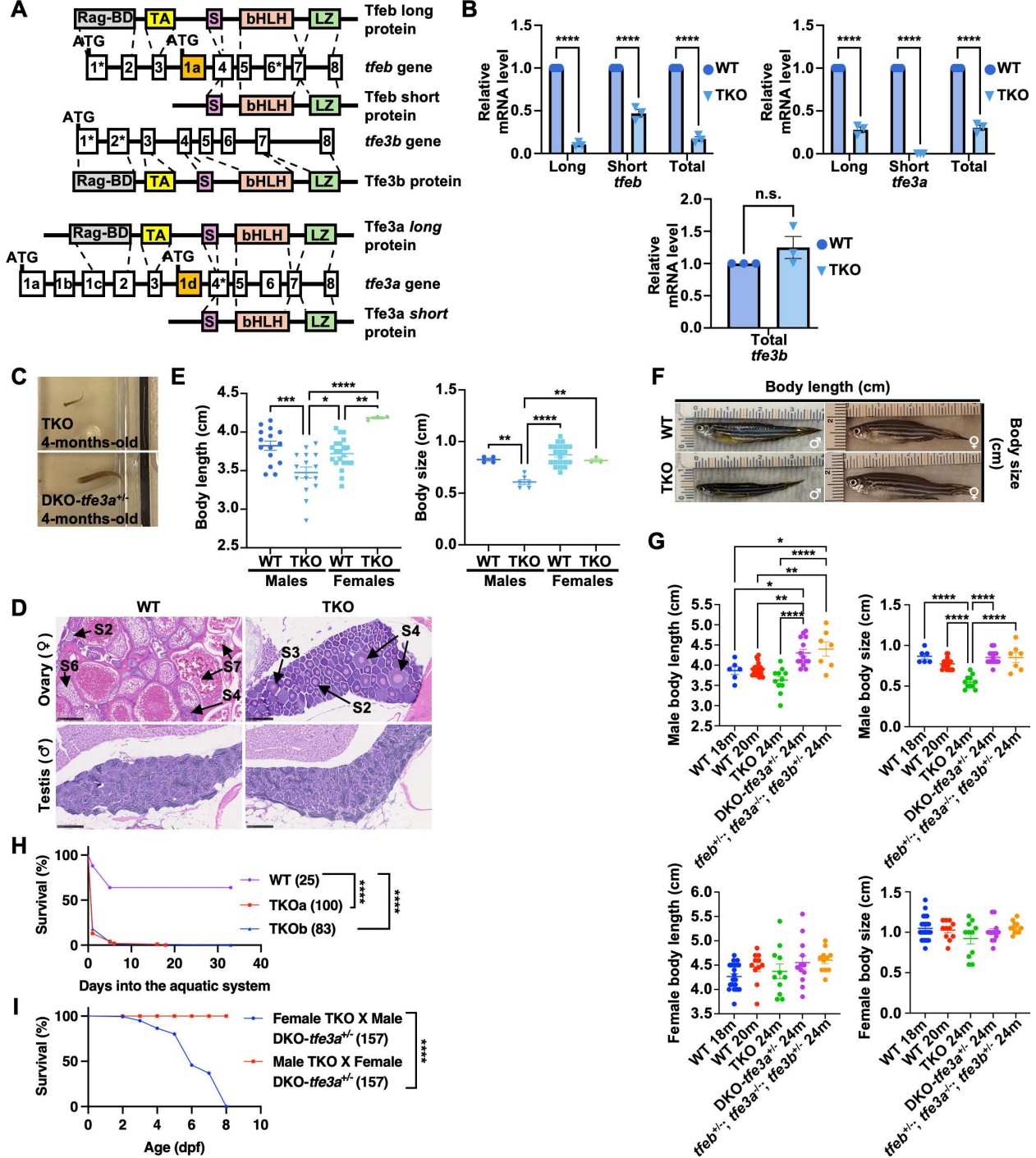

**Fig 1. Generation of zebrafish *tfeb*, *tfe3a* and *tfe3b* triple knockout mutants. (A)** Genomic coding structure of the zebrafish *tfeb*, *tfe3a* and *tfe3b* genes and schematic illustration of the major structural domains of their different forms. The coding exons targeted in genome-editing using CRISPR/Cas9 have been marked with asterisks, alternative first coding exons are highlighted in orange. Exons/introns are not drawn to the scale. RAG-BD, RAG binding domain; TA, transactivation domain; S, 14-3-3 binding motif; bHLH, beta Helix-Loop-Helix; LZ, leucine zipper domain. **(B)** Real-time qPCR analysis showing reduction of *tfeb*, *tfe3a* and *tfe3b* expression in 2 dpf TKO embryos. The data represent means ± SEM, n = 3 independent experiments. Statistical significance was determined by using two-way ANOVA with Sidak's multiple comparisons. n.s., not significant, **** < 0.0001. **(C)** Representative pictures of a 4-months-old TKO fish (top) and its sibling fish (bottom) in a DC-96 genotyping tank. **(D)** Comparison of H&E-stained longitudinal sections of WT and TKO 4-months-old fish embedded in paraffin. (Top) Representative pictures of the ovary of TKO small adult female fish showing the lack

of mature oocytes in the TKO ovaries. Different stages of primary oocytes (S2, S3, S4, S6 and S7) are indicated. S2, S3 represent the previtellogenic stages; S6 and S7 are the last stages of oocytes maturation. (Bottom) Representative pictures of testis of 4-months-old control and TKO small adult male fish. Scale bars, 250 μm. **(E)** Comparison of body length (left) and body size (right) between 12-months-old WT and TKO male and female fish. The data represent means ± SEM. Statistical significance was determined by Welch's two-tailed t-test. * < 0.05, ** < 0.01, *** < 0.001 and **** < 0.0001. **(F)** Representative pictures of 12-months-old WT and TKO fish used for the quantification in (E). **(G)** Comparison of body length and size among different males (top panels) and female (bottom panels) adult fish. WT used are 18- and 20-months-old, TKO and their siblings are 24-months-old. The data represent means ± SEM. Statistical significance was calculated using ordinary one-way ANOVA with Tukey's multiple comparison test, with a single pooled variance. * < 0.05, ** < 0.01, *** < 0.001 and **** < 0.0001. **(H)** Kaplan-Meier curves showing the percentage of survival (y axis) of WT and TKO embryos (TKOa and TKOb) through time (x axis) while growing into the aquatic system from 6 dpf (time point 0). Numbers in parenthesis indicate the number of embryos used. (Log-rank (Mantel-Cox) test: **** < 0.0001). **(I)** Representative Kaplan-Meier curves showing the percentage of survival (y axis) of the embryos from two different crosses through time (x axis). Numbers in parenthesis indicate the number of embryos used. (Log-rank (Mantel-Cox) test: **** < 0.0001).

TKO female fish seemed to partially recover their size through time, TKO male fish were never able to fully recover their size even after 1 year of development (Fig 1E and 1F). To further sustain that observation, we compared the size of 24-months-old TKO males and their siblings (tfeb^-/-; tfe3a^+/-; tfe3b^-/-, indicated as DKO-tfe3a^+/- and tfeb^+/-; tfe3a^-/-; tfe3b^+/- fish) with 18- and 20-months-old wild-type (WT) fish (Fig 1G). Only 2-year-old TKO males presented both reduced body length and size compared to all the other classes (Fig 1G).

Notably, embryos obtained from an intercross of TKO adults did not reach adulthood (3 months of age) and started to die around 4–6 dpf (Fig 1H). We observed almost a complete lethality by 8–10 dpf, with only 1 larva over the 183 tested that was able to pass 1 month of development (Fig 1H). As previously described [31], tfeb, tfe3a and tfe3b transcripts are maternally inherited by the embryos. Therefore, we wondered whether the specific lack of maternal tfe3a transcripts might affect the survival of the TKO larvae. Interestingly, crosses between DKO-tfe3a^+/- females (tfeb^-/-, tfe3a^+/-, tfe3b^-/-), which contain tfe3a maternal RNA in oocytes, and TKO males (tfeb^-/-, tfe3a^-/-, tfe3b^-/-) survived to adulthood. In contrast, crosses between TKOs females and male DKO-tfe3a^+/- males died by 8 dpf (Fig 1I). Quantitative RT-PCR analysis showed that only the tfeb and tfe3a long forms are maternally inherited and their accumulation is reduced in TKO oocytes, probably due to a partial degradation induced by NMD (S1C and S1D Fig). S1 Table summarizes the different survival phenotypes/genotype observed.

TFEB and TFE3 mediate upregulation of multiple lysosomal and autophagy genes in mammals, allowing cellular response to a wide variety of stress conditions, including starvation [6], oxidative stress [9], energy homeostasis [25], DNA damage [34], and inflammation [24], among others. To confirm the disruption of zebrafish tfeb, tfe3a and tfe3b function in TKO mutant embryos, we performed qPCR analysis on mRNA extracted from whole embryos, and measured expression of genes involved in some of the afore mentioned processes. As expected, we observed a very significant reduction in the expression of multiple genes involved in lysosomal biogenesis and/or function (S1E and S1F Fig).

Given the severe lethality observed in TKO embryos after ~5 dpf, we also checked expression of apoptotic markers. qPCR analysis at 4 dpf showed a significant increase in the expression of tp53 and some of its target genes, including p21(cdkn1a), ccng1, Δ113tp53, mdm2, baxa and rps27l, in TKO embryos (S1G Fig). To confirm the presence of apoptotic cells, we performed whole mount TUNEL assay staining in TKO and control embryos at different developmental stages (S1H and S1I Fig). The analysis showed the presence of TUNEL^+ cells in specific areas of the embryos. Compared to WT, TKO embryos showed increased TUNEL positivity in retina and brain at 3 and 4 dpf, while at later stages, TUNEL^+ cells were mostly observed in the eye and the pectoral fins of TKO samples (S1H and S1I Fig).

Overall, our data show that the simultaneous lack of all the forms of tfeb, tfe3a and tfe3b transcripts is embryonically lethal, impairing lysosomal activity and inducing apoptosis mainly in neural tissues, such as retina and optic tectum, during embryo development. Maternal tfe3a long forms presented a pivotal role for embryo survival when the other genes/forms were knocked-out, and zygotic tfe3a transcripts (long and short) were not able to compensate for them. Of note, although sufficient for embryo survival, larvae deprived of zygotic tfe3a transcripts presented a strong reduction of their size that persisted in adult male fish.

## Zebrafish *tfeb, tfe3a, tfe3b* triple knock-out embryos present liver and pancreas defects

To further study the overall effect of *tfeb* and *tfe3* depletion on embryo development, we performed proteomic analysis of 5 dpf whole embryos. Among the 7652 proteins analyzed, we found that 856 proteins were differentially expressed in TKO vs WT (fold change >1.5, p<0.05); among them, 436 and 420 were down- and up-regulated, respectively (Figs 2A and S2A, and S2 Table). Interestingly, gene ontology (GO) Molecular Function analysis of the down-regulated proteins showed an enrichment of terms involved in peptidase activity, in particular of those known to be stored in pancreatic zymogen granules [35], such as serine-type endopeptidases (Cela1.6, Zgc:112160, Prss1, Cela1.4, Cela1.1, Ela2l, Ctrl, Prss59.2 Prss59.1, Ctrb1 and Ela3l), alpha-amylases (Zgc:92137 and Amy2a) and carboxypeptidases (Cpa5, Cpa4, Cpb1, Cpa1) (Fig 2B and S3 Table). Moreover, through manual annotation of the dataset, we were able to find two more down-regulated proteins (Cd63 and Cuzd1.1), not picked up by the GO analysis, previously associated to pancreatic zymogen granules [35] (S3 Table). GO analysis using KEGG pathways and a human dataset showed similar results, plus an enrichment of lysosomal-related terms (Fig 2C), partially confirming the results observed in the qPCR analysis (S1E and S1F Fig). Among the up-regulated proteins, the GO analysis using KEGG pathways revealed the enrichment of proteins involved in metabolic activities such as oxidative phosphorylation, TCA cycle, amino acid metabolism and fatty acid catabolism (S2B Fig and S3 Table). Moreover, the GO Molecular Function analysis (nutrient reservoir activity and lipid transporter activity terms) showed that most of the vitellogenins (glycoproteins accumulated in the oocytes of adult females and main component of the yolk) [36], were strongly enriched and upregulated in TKOs compared to controls (S2C and S2D Fig and S3 Table, respectively).

Whole mount in situ hybridization (WISH) analysis showed that *tfe3a* transcripts (long and short) were expressed in neural tissues and gastro-intestinal organs during normal development (Fig 2D); moreover, TKO embryos presented altered expression of both, pancreatic (*try* and *ins*) and hepatic markers (*fabp10a*) (Fig 2E). The reduction of *trypsin* expression in TKO pancreas, especially in the most central part of the organ, was very substantial (arrowheads in Fig 2E). H&E analysis of paraffin sections of TKO embryos at the same developmental stages confirmed the presence of morphological abnormalities in both the organs (Figs 2F and S2E). In TKO embryos, the pancreas presented reduced size and decreased eosinophilic staining at the most apical portion of the acinar cells, rich in zymogen granules (Fig 2F). The only islet present at these developmental stages appeared slightly reduced, but overall, morphologically comparable to the controls (Figs 2E, 2F and S2E). Moreover, WISHs and analysis of serial sections of the trunk region of the embryos at 5 dpf confirmed reduced yolk adsorption in TKO embryos (Figs 2E, 2F and S2F-I), presence of discontinuities in the exocrine pancreas tissue (arrows in S2H and S2I Fig), and reduction of the central part of the organ, as previously observed in WISH analysis (black arrowheads in Figs 2E and S2I). Similarly to pancreas, TKO livers presented reduced size compared to control embryos with reduced eosin staining and more dispersed nuclei, suggesting decreased cellular density (Figs 2E, 2F and S2F-I).

Overall, our proteomic and histological data showed that TKO embryos presented severe developmental defects in gastro-intestinal organs, in particular pancreas and liver, which might contribute to the observed increased lethality.

## Characterization of pancreatic alterations in TKO embryos

To further characterized the alterations observed in the liver and pancreas of TKO embryos, we moved the TKO fish to the *2CLIP;Tg(ptf1a:EGFP)^jh1* transgenic background [37]. This transgenic line expresses GFP in pancreatic acinar cells and multiple neuronal regions (retina, brain and spinal cord), as well as dsRED in hepatocytes and pancreatic beta cells, thus allowing the study of pancreas and liver embryonic development in real time.

By using confocal microscopy, we were able to visualize an array of morphological defects in the pancreas of the TKO embryos from ~4 dpf, with a complete penetrance in the population starting from 5 dpf (Figs 3A, 3B, S3A and S3B). As previously observed in WISH experiments and histological analysis (Figs 2E and S2F-I), our confocal analysis confirmed

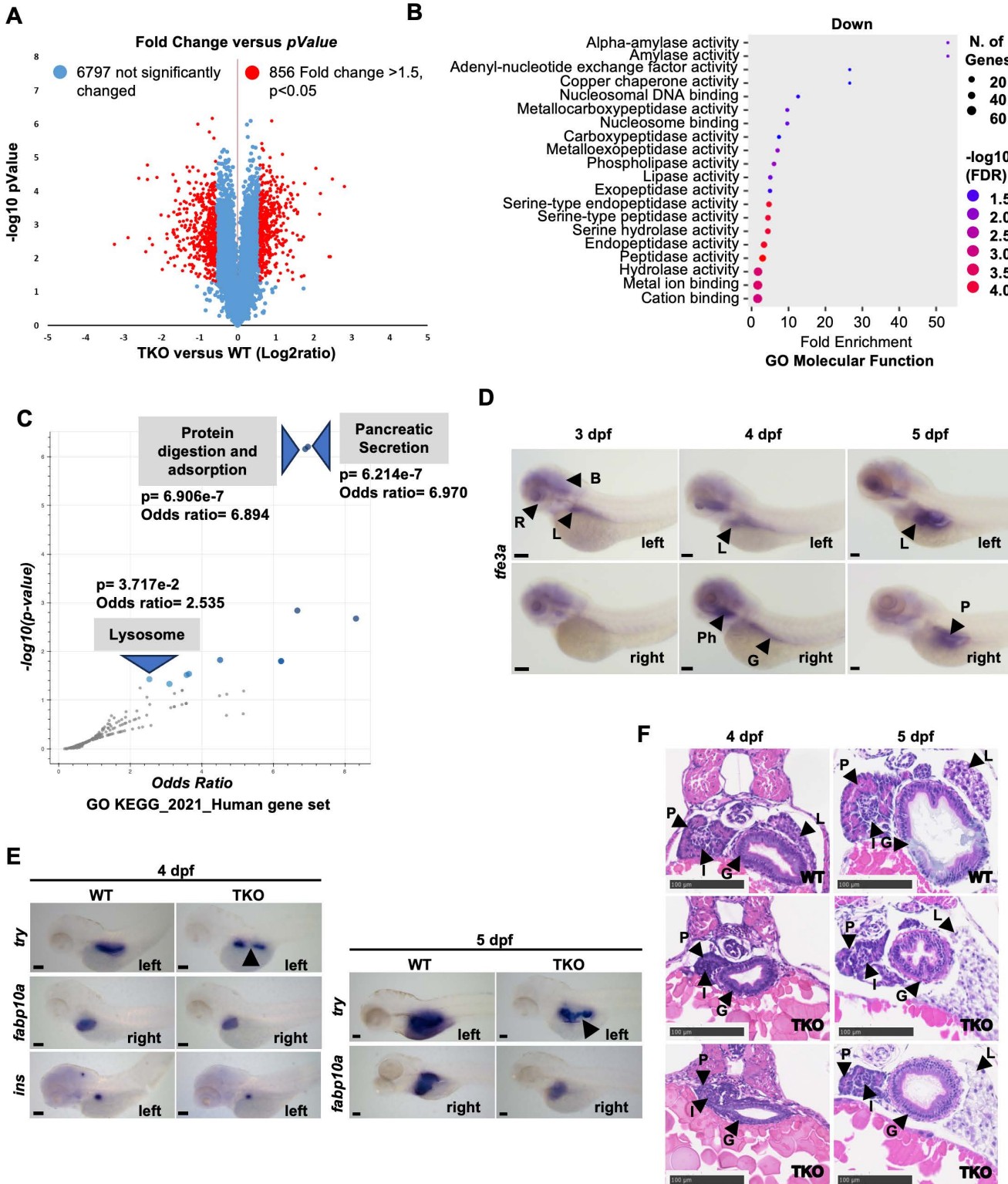

**Fig 2. Proteomic and phenotypical analysis of 5 dpf WT and TKO embryos. (A)** Volcano plot showing the fold change of 7652 protein abundance between TKO zebrafish group and WT group (n = 3). The x-axis represents the log2 of fold changes, and the y-axis represents the statistically significant p-value (-log10 of p-value, n = 3). Blue dots represent 6797 proteins fold change <1.5, red dots are 856 differentially expressed proteins with a fold

change >1.5, p<0.05. **(B)** ShinyGO Molecular Function term enrichment of the proteins significantly downregulated in TKO mutants compared to WT siblings. FDR, false discovery rate. **(C)** Volcano plot of terms from the KEGG_2021_Human gene set using Enrichr. Each point represents a single term, plotted by the corresponding odds ratio (x-position) and -log10(p-value) (y-position). Blue points represent significant terms (p-value<0.05); smaller gray points represent non-significant terms. **(D)** WISH analysis with an antisense probe against *tfe3a* transcripts during WT embryo development. Top and bottom panels show left and right sides of the embryos, respectively. Anterior to the left. B, brain; G, gut; L, liver; P, pancreas; Ph, pharynx; R, retina. Scale bars, 100 µm. **(E)** WISH analysis with an antisense probe for pancreatic (*try* and *ins*) and hepatic (*fabp10a*) markers in WT and TKO at 4 and 5 dpf. Arrowheads point to strong reduction of *try* signal in TKO embryos. Lateral positions, anterior to the left. Scale bars, 100 µm. **(F)** H&E staining of 5 µm transversal sections showing the gastro-intestinal organs in the trunk of paraffin embedded 4 and 5 dpf WT and TKO embryos. G, gut; I, islet; L, liver; P, exocrine pancreas. Scale bars, 100 µm.

the presence of discontinuities in the exocrine pancreas tissue and the reduction of the central part of the organ. Furthermore, we observed a very peculiar defect in the orientation of the pancreas in a subset of the TKO population (S3C-E Fig). Overall, TKO embryos presented higher penetrance of laterality defects compared to control WT embryos, with 6.9% (60/867) and 0.9% (4/426) of the respective populations showing *situs inversus* of the visceral organs. Notably, 24% (209/867) of the TKO embryos presented also an incorrect orientation of the pancreas (S3C-E Fig) regardless of the presence of the *situs inversus* (S3C Fig). The same phenotype was observed in the control population as well, although at a very low frequency (0.5%, 2/426 control embryos) (S3E Fig).

Further studies using semithin sections and toluidine blue staining revealed a strong reduction of the staining of zymogen granules (ZGs) in the acinar cells of the TKO embryos (red area in S3F Fig). This observation was confirmed by electron microscopy (EM) analysis (Fig 3C). At sub-cellular level, TKO pancreatic acinar cells presented a significant reduction in the size of ZGs, as well as a higher percentage of abnormal granules/area (Fig 3D and 3E, respectively), corroborating the data from the proteomic analysis (Fig 2B and 2C) and suggesting a reduced functionality of the organ. Despite the higher percentage of abnormal ZGs in TKO EM sections, we only observed a few fused with functional lysosomes (Fig 3C, white arrows). This suggests that the impairment of lysosomal and/or autophagic activity caused by the lack of *tfeb*, *tfe3a* and *tfe3b* may prevent the removal of damaged/fragile zymogen granules, leading to intracellular activation of pancreatic enzymes and eventually to pancreatitis [38–40]. To further test this hypothesis, we performed histological analysis on 4-month-old WT and TKO adults derived from an intercross of DKO-*tfe3a*+/- fish, which can reach adulthood (Fig 3F). Histological analysis highlighted defects in the exocrine pancreas of TKO adults. Independently of the size of the TKO adult fish, we observed some early signs of pancreatic inflammation [41,42], such as generalized edema and a high level of eosin-negative zymogen granules in acinar cells (Fig 3F). Our analysis did not show any major signs of fibrosis or massive infiltrations of immune cells, as observed in other models of pancreatitis [41], suggesting the presence of mild defects, more similar to those previously observed in an acinar cell-specific tfeb KO murine model [40]. The endocrine component of the organ did not present major defects, suggesting a specific effect of the lack of *tfeb*, *tfe3a* and *tfe3b* on the exocrine component of the pancreas, at least in the adult mutants (Fig 3F). Further characterization on older fish (7 months) showed a reduction of ZGs toluidine staining in TKO acinar cells (S3G Fig). EM analysis confirmed a strong accumulation of abnormal ZGs in TKOs, many of them presenting damaged or disrupted membranes, suggesting the potential release of pancreatic enzymes (Fig 3G). It was also clear the presence of dilated endoplasmic reticulum and nuclear envelop that may indicate increased ER stress (Fig 3G).

## Single-cell gene expression profile of WT and TKO acinar pancreatic cells at 4 dpf

To comprehensively characterize the transcriptional landscape of the TKO embryos, we performed single-cell RNA sequencing (scRNA-seq) of dsRED+- and GFP+-positive cells isolated from our 2CLIP lines. We decided to use embryos at 4 dpf, 1 day before the stage used for proteomic analysis, because we reasoned that transcriptional variations usually precede changes at protein level. The experimental workflow used for the scRNA-seq experiments is shown in S4A Fig. Since GFP is expressed in several tissues (pancreas, retina, brain and spinal cord), we first enriched

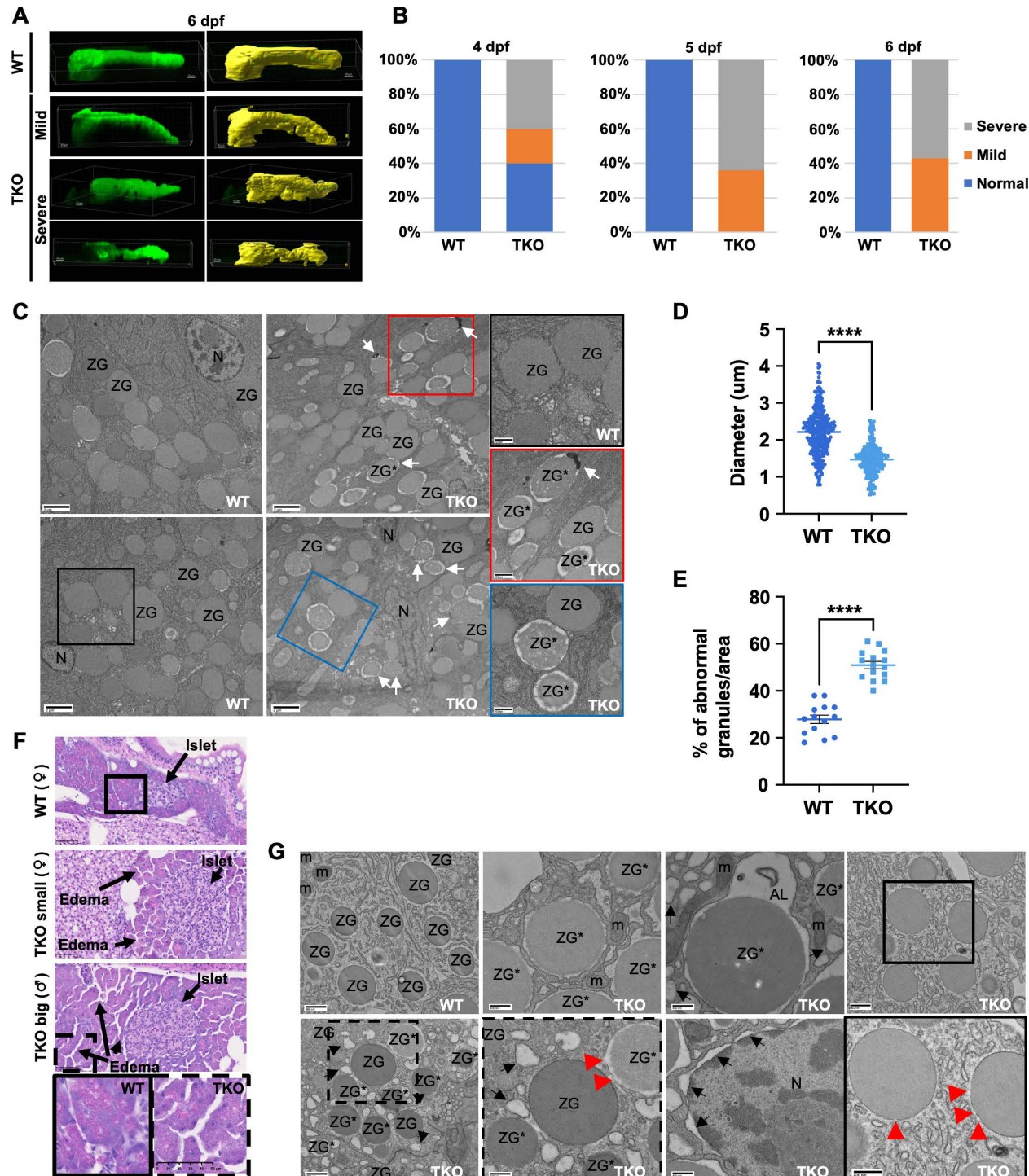

**Fig 3. Pancreatic defects observed in TKO embryos and adult fish. (A)** Representative max-projections (green) and Imaris 3D reconstructions (yellow) of confocal Z-stacks of 2CLIP WT and TKO embryos at 6 dpf showing the different classes of phenotypes of the mutant embryos. Dorsal views, anterior to the left. Scale bars, 30 μm. **(B)** Quantification of the % of pancreas phenotypes observed in WT and TKO embryo populations during development from 4 to 6 dpf. **(C)** Representative EM pictures comparing acinar cells in WT and TKO 5 dpf embryos. White arrows: zymogen granules in autolysomes. Scale bars, 2 μm. Inset scale bars: 0.8 μm. **(D-E)** Quantification of zymogen granules size (D) and quality (E) in WT and TKO embryos at 5 dpf. The data represent means±SEM. Statistical significance was determined by unpaired two-tailed t-test. ****<0.0001. **(F)** H&E histological analysis of 4

months old WT and TKO adult fish showing mild signs of pancreatic inflammation (edema, eosin-neg zymogen granules). Insets are showing the edema and the lack of eosin positivity in the TKO mutant pancreas more in detail. Scale bars, 50 μm. **(G)** Representative EM images of pancreatic tissues from adult WT and TKO fish. Scale bars: 800 and 400 nm. Black arrows mark dilated endoplasmic reticulum and nuclear envelop; red arrowheads label ruptures in the zymogen granule membrane. AL, autolysosome; m, mitochondria; N, nucleus; ZG, zymogen granule; ZG*, abnormal granules.

the pancreatic cell population by manually performing a resection of the most anterior trunk region from ~150 WT and TKO embryos (S4A Fig). After cellular dissociation, cells were stained to detect live and dead cells, then sorted, and processed for bioinformatic analysis (S4A-E Fig). The transcriptional profiles of WT and TKO GFP+ cells were analyzed by scRNA-seq, and an independent Uniform Manifold Approximation and Projection (UMAP) was generated for each dataset (S4F Fig), showing 14 different clusters in both datasets. The UMAP of the merged WT and TKO cells defined 17 cell clusters (S4G Fig). Cell identities were assigned to each cluster based on well-established lineage markers identified as clusters signatures (S4 Table). Then all the neural (Clusters 0, 1, 4, 10–16) and liver (Cluster 3) clusters were removed from the analysis to focus on pancreatic cells (S4H Fig). A new UMAP was generated using the remaining clusters (S4I Fig) and their expression profiles were used to assign the identity of each different new cluster, when possible (Figs 4A, 4B and S4J, and S4 Table). We were able to assign specific identity to most of the clusters, except for cluster 3, which probably represented a mix of different cell types (Fig 4A, left panel). Analysis of cell genotype in the different clusters showed that most (clusters 2–10) were a mix of WT and TKO cells, with a relatively similar expression profile (Fig 4A, right panel).

Clusters 0, 1 and 9 corresponded to pancreatic acinar cells, with cluster 9 representing a mixed group of mitotic WT and TKO cells (S4J Fig). Importantly, clusters 0 and 1, corresponding to TKO and WT acinar cells, respectively, presented very different expression profiles (Fig 4A and S4 Table), so we focused our attention on these two clusters. Initially we wondered whether pancreatic enzymes may be downregulated at transcriptional level, since this could explain their reduced expression in the proteomic dataset at 5 dpf (Fig 2B and 2C). However, the scRNAseq data showed that the expression of most of those genes appeared unchanged in TKOs or, in some cases, it was slightly upregulated compared to the controls (Fig 4C). These results suggest that the reduction of the pancreatic enzymes is not the result of alterations in their transcriptional levels but might be due, instead, to the increased fragility of the TKO zymogen granules observed later during development (Fig 3C-E) and in the adults (Fig 3F-G).

In agreement with this idea, we found that the expression of critical autophagy modulators, such as *sqstm1*, *optn*, *ubc*, *ulk1a*, *ulk1B*, and *clec16a*, was strongly reduced in TKO acinar cells (Fig 4D). Particularly relevant may be the reduced expression of *sqstm1*, which have been previously implicated in elimination of defective zymogen granules or zymophagy [38], as well as *vmp1*, a protein implicated in autophagosome formation that when depleted in mouse acinar cells, causes acute pancreatitis [43]. Expression of several markers of lysosomal function (*cd63*, *ctsla*, *catsd*), lysosomal repair (*chmp4ba*, *chp4bb*) and iron homeostasis (*fthl27*, *fthl30*, and *fth31*) was also significantly decreased in TKOs (Fig 4E).

When we next analyzed other stress pathways, we found a slight upregulation in the expression of several ER stress markers, including *xbp1*, *atf6*, *eif2ak3* (perk) and *hsp90b1* (grp94) in TKO acinar cells. However, the levels of *atf4* (*atf4a* and *atf4b*) and several *atf4* targets (*atf3*, *ddit3*) were decreased (Fig 4F), which is consistent with previous reports suggesting that in mammals, TFEB and TFE3 are required for an efficient ATF4-mediated response [44]. This suggests the presence of ER stress that cannot be properly resolved. In contrast, we did not observe a general increase in the expression of oxidative stress related genes, with only *prdx3* and *prdx6* showing a very minor upregulation (Fig 4G). Surprisingly, *nos1* (the neuronal-specific form of nitric oxide synthase) appeared strongly upregulated in TKO cells (Fig 4G). Expression of *mdm2*, a critical regulator of t*p53*, was also clearly increased in TKOs (Fig 4G). Finally, we found altered expression of some transcriptional regulators. The expression of several well-known TFEB targets, such as *bhlhe40*, *bhlhe41*, *jun*, and *tgf1* was severely reduced in TKOs, while others, including *ptf1a*, *her6*, and *her9*, showed upregulation (Fig 4H).

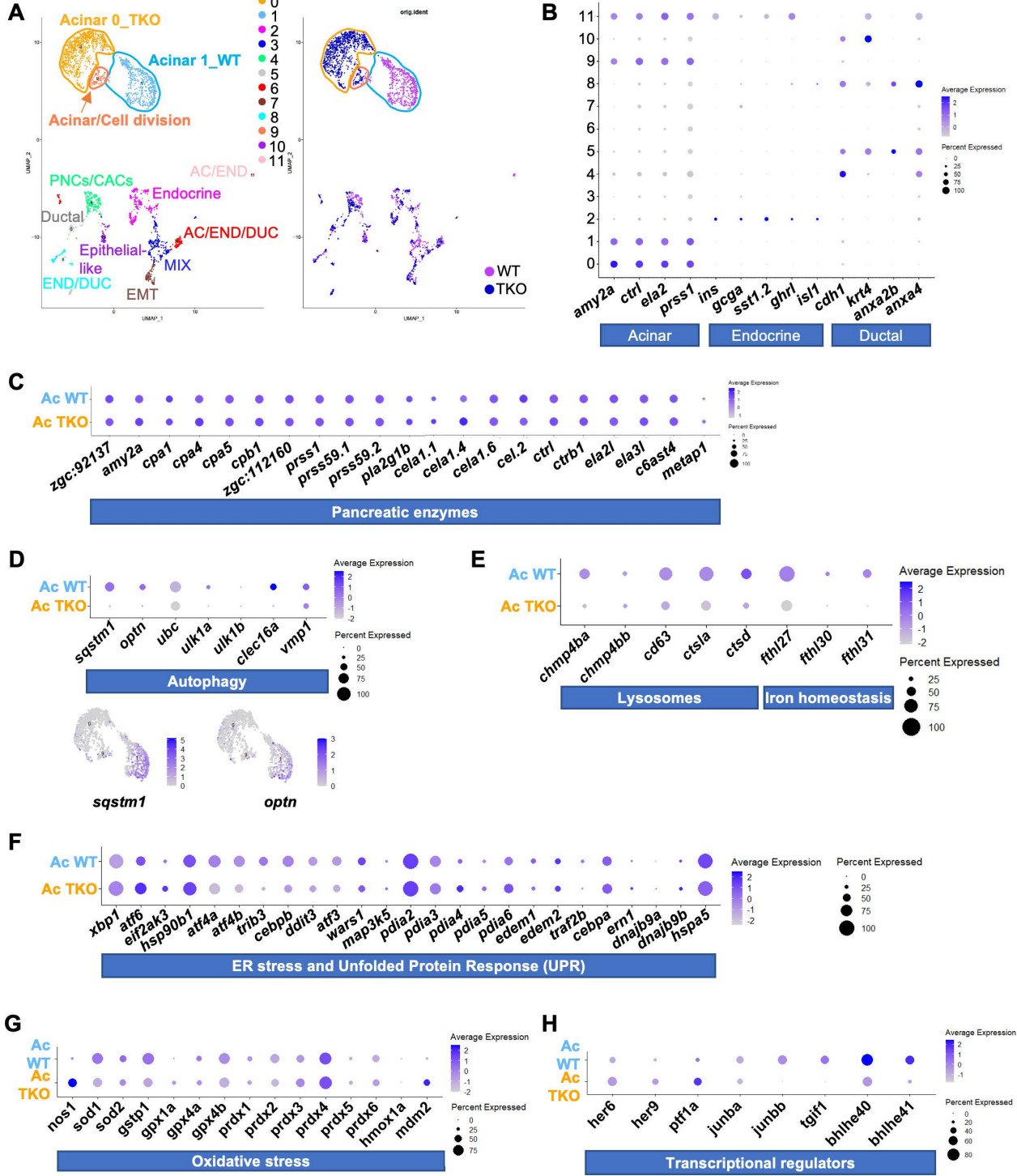

**Fig 4. Single cell RNA-sequencing of 4 dpf WT and TKO GFP⁺ cells. (A)** (Left) UMAP showing clustering of merged WT and TKO GFP⁺ datasets. Each cluster identity has been annotated in the plot. Acinar cell clusters have been highlighted. (Right) UMAP displaying the projection of the two genotypes (WT and TKO) merged for the analysis. **(B)** Dot plots showing the expression of acinar, endocrine and ductal gene markers in merged WT and TKO datasets as shown in (A). **(C)** Dot plot showing the expression of the pancreatic proteins found downregulated in the proteomic analysis of 5 dpf WT and TKO embryos. **(D)** (Top) Dot plot showing expression of genes involved in autophagy in TKO and WT acinar cell (Ac) clusters. (Bottom) feature plots depicting the expression level of specific autophagic markers in acinar cell clusters as shown in (A) (purple is high, gray is low). **(E-G)** Dot plot showing

expression in TKO and WT acinar cell (Ac) clusters of genes involved in different cellular processes: lysosomes and iron homeostasis (E), ER stress and UPR (F) and oxidative stress (G). **(H)** Dot plot showing examples of genes differentially expressed in TKO and WT acinar cells at 4 dpf. In each dot plot, the size of the dots encodes the percentage of cells within a class, the color indicates the average level of expression (purple is high, grey is low). Graphics in S4A Fig were created with BioRender.com.

Overall, our data suggest that defective autophagy and lysosomal function results in increased ER stress and accumulation of damage granules, leading to the disruption of acinar cells in TKOs.

## Lack of zebrafish *tfeb*, *tfe3a* and *tfe3b* induces liver and gut defects

The pattern expression analysis of the *tfe3a* gene showed specific expression in liver and gut around 3–5 dpf (Fig 2D) and the expression of specific liver markers appeared reduced in TKO embryos at 4 dpf (Fig 2E). In addition, histological analysis suggested morphological differences in TKO embryos, in particular a reduction of the liver cell density and of the size of the gut in TKO adults (Figs 2F, S2F-I and S3F). Given the lack of TUNEL$^+$ cells in the TKO embryos in these organs (S1I Fig), we wondered whether cell cycle defects might be present. To test this hypothesis, we took advantage of the dual Fucci system [45] to visualize the nuclei of cells in $G_0/G_1$ and $S/G_2/M$ phases in red and blue, respectively, in WT and TKO embryos. In agreement with previous reports [46,47], confocal analysis of FUCCI control embryos showed active proliferation in liver around 2–3 dpf, with a significant increase in the number of cerulean$^+$ nuclei, and then reached tissues homeostasis around 7 dpf, with most cells in $G_0/G_1$ phase (S5A and S5B Fig). On average, most cells divided every hour in control embryos (S5B Fig). Conversely, analysis of confocal movies of the TKO embryos at different stages showed multiple examples of cerulean$^+$ nuclei that did not divide, even after more than 3 hours of observation (Fig 5A). The block of some hepatic and gut epithelial cells in $S/G_2/M$ may explain the size reduction of these organs found later during development (Figs 2E, 2F, S2F-I and S3F). To further sustain this hypothesis, we compared the number of quiescent cells (mCherry$^+$ nuclei) in similar areas of the left lobe of the liver between WT and TKO embryos at 6.5 dpf, when the tissue should have reached the cellular homeostasis. The analysis confirmed a significantly reduced number of mCherry$^+$ nuclei per volume of tissue in TKO mutant compared to the controls (Fig 5B).

To highlight possible subcellular defects, we performed EM analysis of embryonic liver and gut epithelium (Fig 5C and 5F, respectively). Interestingly, we observed hepatic alterations with different levels of severity in TKO embryos (n = 3 animals): from an increased number of peroxisomes and their close association to mitochondria (Fig 5C), to more severe defects including distressed mitochondria showing intracristal swelling and blistering of the external membrane, as well as accumulation of very small lipid droplets (Fig 5C, insets). In addition, all the TKO liver sections showed decreased glycogen accumulation inside hepatocytes (Fig 5C). This initial observation was confirmed by calculating the area occupied by glycogen lacunae in WT and TKO hepatocytes at 5 dpf (Fig 5D) and by performing quantification of the total content of embryonic glycogen at 4 and 5 dpf (Fig 5E). Reduction of glycogen storage and increased peroxisome numbers were also observed in TKO adult hepatocytes (S5C-E Fig). Our EM analysis also identified mild defects in the gut epithelium of the TKO embryos at 5 dpf (Fig 5F and 5G). Although the morphology of the gut enterocytes appeared comparable to controls (Fig 5F), quantification of the microvilli on their surface showed a statistically significant reduction of their density (Fig 5G), suggesting a possible reduction of the adsorption ability of the intestine in TKO embryos.

Proteomic analysis of livers extracted from WT and TKO adults showed a relative limited number of differentially expressed proteins (245 out of 7860) (S5F Fig and S5 Table). However, GO KEGG analysis of downregulated and upregulated proteins revealed alterations in several metabolic pathways in TKO livers (S5G and S5H Fig and S6 Table). One of the most upregulated proteins in TKO liver was glucose-6-phosphate dehydrogenase (G6pd), the rate-controlling enzyme of the pentose phosphate pathway [48] (S5 and S6 Tables). Catalase (Cat), a specific peroxisomal marker, was also significantly upregulated, which agrees with the increased number of peroxisomes observed in the EM sections (S5 and S6 Tables, S5D and S5E Fig).

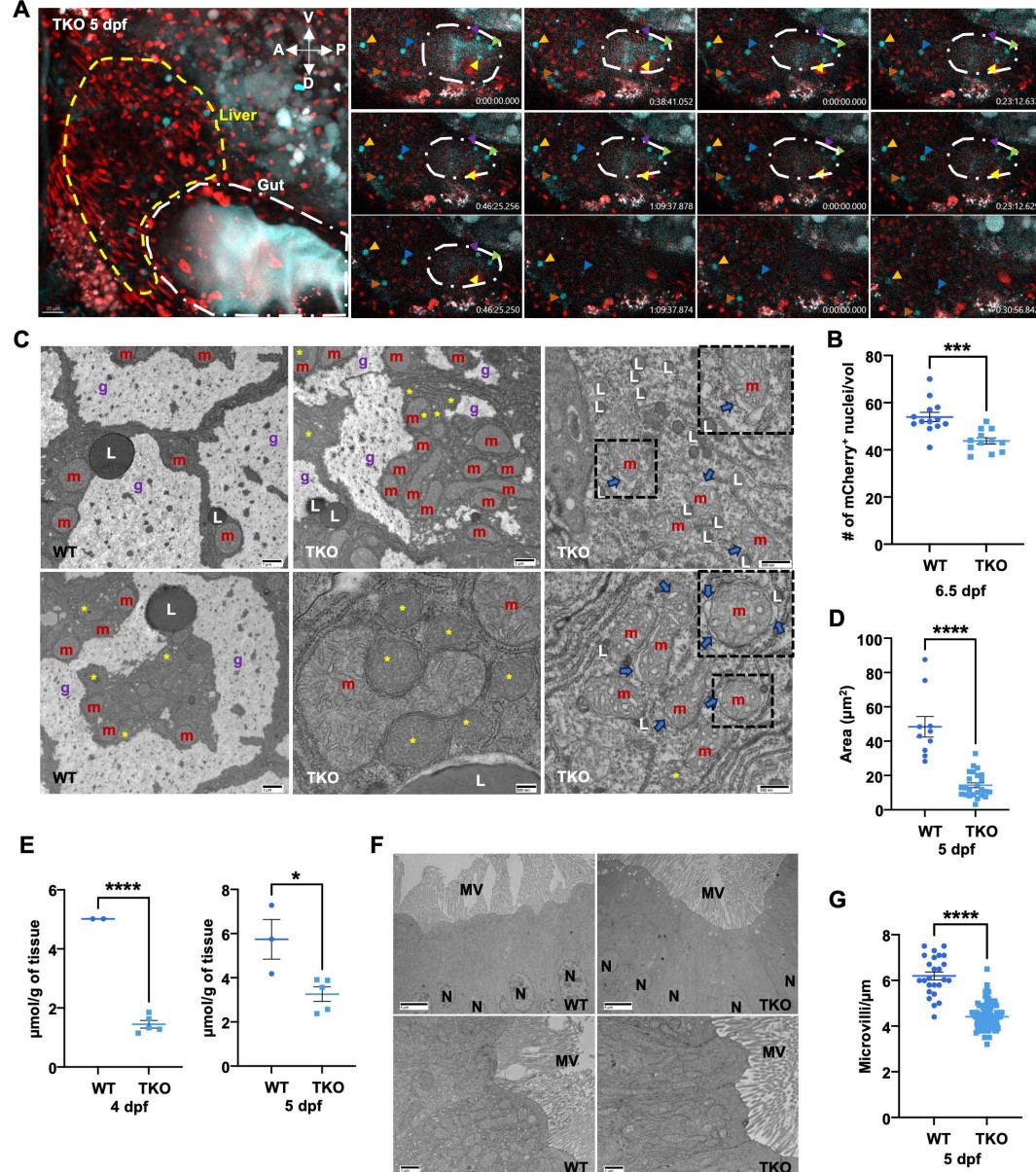

**Fig 5. Liver and gut defects observed in TKO embryos. (A)** (Left) Representative max-projection of confocal Z-stacks of a FUCCI TKO embryo at 5 dpf. The dashed yellow and white lines highlight the left lobe liver and the gut, respectively. Lateral views, anterior to the left and dorsal to the bottom. Scale bar, 20 μm. (Right) Pictures of the same single plane over time from each confocal Z-stack. For clarity, only the gut region is highlighted with a white long-dashed dot line. Arrowheads point at the same cerulean+ cells over time. Numbers indicate the time of each frame; each hour the timer is restarting from zero. **(B)** Quantification of the number of $G_0/G_1$ nuclei per volume ($39\mu m^3$) in the left lobe of the liver of WT and TKO at 6.5 dpf. **(C)** Representative EM pictures showing subcellular organization of hepatocytes in 5 dpf WT and TKO embryos. Different grades of severity have been observed in TKO hepatocytes. Dashed black rectangles highlight the insets. m, mitochondria, L, lipid droplets, g, glycogen lacunae, yellow asterisks mark the peroxisomes. Blue arrows point to mitochondria showing signs of distress (intracristal swelling and blistering of the external membrane). Specific scale bars are indicated for each picture. **(D-E)** Quantitative analysis of glycogen content in WT and TKO at different stages. (D) Analysis performed using ImageJ (Fiji) on EM images at the same magnification from WT and TKO at 5dpf. **(E)** Glucose content (μmol/grams of embryonic tissue) quantification after amyloglucosidase digestion in WT and TKO embryos at 4 and 5 dpf. **(F)** Representative EM pictures comparing gut epithelial cells in WT and TKO 5 dpf embryos. Scale bars, 4 μm (top) and 1 μm (bottom). N, nucleus, MV, microvilli. **(G)** Quantification of the microvilli density (# of microvilli/μm) in 5 dpf WT and TKO gut epithelial cells. The data represent means ± SEM. Statistical significance was determined by unpaired two-tailed t-tests. * < 0.05, *** < 0.001 and **** < 0.0001.

## Single-cell gene expression profile of WT and TKO hepatocytes at 4 dpf

Next, we sorted dsRED+ cells from 2CLIP WT and TKO fish at 4dpf, which correspond to hepatocytes and endocrine pancreatic cells (S6A-C Fig). Since hepatocytes contain a high number of mitochondria (S6D Fig), as previously reported [49], we set a<60% of percent.mt as a threshold to be sure to include the hepatocyte population in the analysis. The transcriptional profiles of WT and TKO dsRED+ cells were analyzed by scRNA-seq and, using the same parameters, an independent UMAP was generated for each genotype (S6E Fig), showing 6 different clusters in both the cases. Cell identities were then assigned to each cluster based on the presence of typical markers identified as clusters signatures (S6F Fig and S7 Table). Both WT and TKO cells presented 4 clusters for hepatocytes (Clusters 0–4 and Clusters 0-1-4-5, respectively), an undefined mixed cluster of cells expressing neuronal markers (Cluster 3), and a cluster of endocrine cells from the pancreas (Cluster 5 and Cluster 2, respectively) (S6E and S6F Fig and S7 Table). Notably, when we compared the mitochondrial QC metric through the different clusters of WT and TKO cells, it appeared that only the hepatocyte clusters were responsible for the observed high percentage of mitochondrial DNA (S6G Fig), excluding the presence of generic stress conditions during the dissociation and sorting that could have affected their overall integrity and survival. To better compare the expression profile of WT and TKO cells we merged the datasets and generated a new UMAP (Fig 6A). The UMAP of the merged WT and TKO cells defined 9 cell clusters (Fig 6A), and their expression profiles were used to assign their identity (S6H Fig and S7 Table). We identified Clusters 0–4 as hepatocytes, Cluster 7 as dividing hepatocytes, and Clusters 6 and 8 as endocrine cells (Fig 6A). As previously observed (Clusters 3 in S6E and S6F Fig), only Cluster 5 likely represented a mix of different cell types and genotypes (Figs 6A and S6H). Based on the genotype, most of the clusters presented a certain degree of cell separation in the UMAP, suggesting a potentially different expression profile (Fig 6A, right panel).

Since we observed cell-cycle defects in TKO hepatocytes *in vivo* during embryo development (Fig 5A), we decided to compare cell-cycle status between different WT and TKO clusters using Seurat CellCycleScoring function (Figs 6B and S6I). As expected, almost all the cells in Cluster 7 were assigned to a proliferative phase of the cycle ($G_2$/M or S phases) (S6I Fig). Overall, WT samples presented a statistically significant increase in the number of cell in $G_2$/M phase (S6I Fig). Conversely, the comparison between TKO Cluster 0 and WT Clusters 1-3-4 revealed increased number of TKO cells in $G_1$ phase and a significant reduction of the proliferating cells, particularly those in the $G_2$/M phase (Fig 6B), confirming the block in S phase of a portion of the hepatocytes observed *in vivo* (Fig 5A and 5B).

To further compare hepatocyte expression profiles between WT and TKO, we modified the resolution parameter and repeated the analysis. The UMAP in Fig 6C shows the resulting organization of hepatocyte clusters. We then performed a DE analysis comparing Cluster 1 (TKO) and Cluster 0 (WT>>TKO cells) (S7 Table). The list of differentially expressed genes was used to perform a GO analysis based on the KEGG database (S6J Fig). The analysis showed a general enrichment of several metabolic pathways, including TCA cycle, oxidative phosphorylation, glyoxylate and dicarboxilate metabolism, amino acid metabolism, and lipid metabolism, among others (S6J Fig). Another enriched category was glutathione metabolism, which was also found in the GO analysis performed on the proteomic data on adult liver (S5H Fig), suggesting higher expression of antioxidant genes in TKOs, possibly due to increased level of oxidative stress. Since we observed a reduction of glycogen accumulation in both TKO embryonic and adult hepatocytes, as well as the presence of stressed mitochondria in embryonic hepatocytes (Figs 5C-E and S5C-E), we focused our attention on specific metabolic pathways (glycolysis, glycogenolysis and oxidative phosphorylation) and expression of antioxidant genes (Fig 6D-F). Instead of limiting the analysis to the known DEGs, we checked expression of those genes included in each corresponding gene sets from the KEGG database in Cluster 1 and Cluster 0 as shown in Fig 6C. Analysis of the glycolysis, gluconeogenesis, and glycogenolysis pathways confirmed a general upregulation of most genes in TKO cells (Fig 6D), further supporting the *in vivo* data regarding the reduction of glycogen in TKO hepatocytes. As shown in Fig 6E, we observed a general upregulation of cellular components of the oxphos genes; however, at the same time, all the mitochondrial genes involved in different complex of the respiratory chain (complex I, complex III, complex IV and complex V) [50]

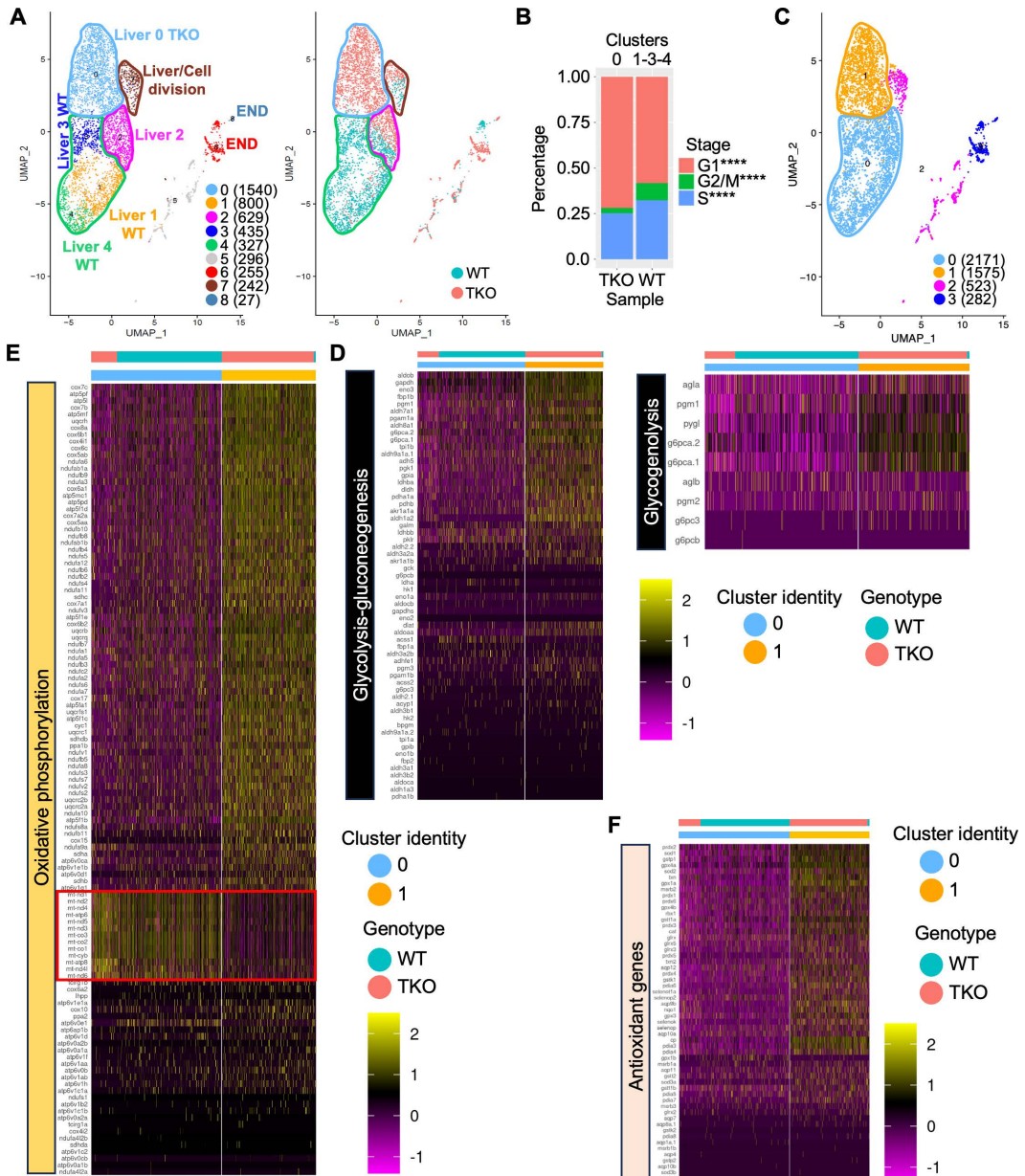

**Fig 6. Single cell RNA-sequencing of 4 dpf WT and TKO dsRED⁺ cells. (A)** (Left) UMAP showing the clustering of merged WT and TKO dsRED⁺ cells. Liver and endocrine pancreatic cell cluster identity has been annotated in the plot. The total number of cells in each cluster in shown in parenthesis. (Right) UMAP displaying the projection of the two genotypes (WT and TKO) merged for the analysis. Major liver clusters have been highlighted. **(B)** Analysis of the cell cycle using Seurat CellCycleScoring function in WT and TKO hepatic cells. Statistical significance was determined using Fisher's exact test. ****<0.0001. **(C-F)** Heat maps showing expression of metabolic genes involved in glycolysis/gluconeogenesis and glycogenolysis (D), oxidative phosphorylation (E), and antioxidant genes (F) at single-cell level in WT and TKO liver cell clusters highlighted in the specific UMAP (C). Numbers in parenthesis indicate the total number of cells for each cluster.

appeared strongly downregulated (red box), suggesting a block of the mitochondrial respiratory chain and ATP production. Increased levels of intracellular ROS can be a direct consequence of a partial/complete block of the mitochondrial respiratory chain [51], so we checked the expression of antioxidant genes in our dataset (Fig 6F). As expected, most of the

genes presented an increased expression in the TKO cells compared to the controls (Fig 6F), suggesting that the abnormal morphology of the mitochondria observed in the TKO embryos at 5 dpf (Fig 5C) might be the direct consequence of a block of the mitochondrial oxidative phosphorylation. GO analysis found an enrichment of peroxisomal and fatty acid degradation genes among the upregulated genes (S6J Fig), confirming our previous EM analysis in both embryonic and adult liver tissues (Figs 5C and S5C-E). Altogether, our data show an overall metabolic rewiring in TKO hepatocytes, which likely represent the cellular attempt to adapt to defective mitochondrial function and increased oxidative stress.

### *tfeb*, *tfe3a* and *tfe3b* triple knock-out embryos are more sensitive to cellular stresses

To test whether the lack of *tfeb*, *tfe3a* and *tfe3b* genes could predispose embryos to a higher sensitivity to stress conditions, we exposed embryos to different environmental stressors. As mentioned earlier, our analysis suggested increased oxidative stress in basal conditions in TKO animals. Therefore, we started by increasing cellular levels of oxidative stress *in vivo* by treating WT and TKO embryos with different doses of $NaAsO_2$ from 1 dpf to 7 dpf, and then we monitored their survival rate [12,52]. WT embryos were not particularly sensitive to the lower dosage treatments; only dosages of 1.5 mM or above for several days of treatment were able to significantly affect their survival (Figs 7A-D and S7A). In contrast, TKO embryos presented a high lethality rate at all the dosages examined (Figs 7A-D and S7B); we did not observe statistically significant differences between 1.5 and 2 mM treatments, suggesting that $NaAsO_2$ reached its plateau of efficacy at 1.5 mM (S7B Fig). The higher sensitivity of the TKO embryos to $NaAsO_2$ exposure, further confirmed the important role of zebrafish *tfeb*, *tfe3a* and *tfe3b* in this specific cellular stress response.

Manual annotation of the 5 dpf proteomic dataset highlighted the presence of differentially expressed proteins known to be involved in protein folding and/or other stress conditions (S7C Fig). For this reason, we decided to use chronic heat shock treatment as an additional environmental stressor and check its effect in survival of TKO embryos (Fig 7E-I). Since it has been shown that zebrafish embryos are most susceptible to heat shock treatments during the first cycles of cell divisions (early cleavage stages), acquiring increased resistance to heat as they progress through the development (blastula, gastrula and further stages) [53], we decided to use the 30–50% epiboly stage, corresponding to the beginning of the gastrulation, as the earliest stage for the heat shock treatments (Fig 7F) [54]. We also tested different start points, exposing embryos from the same batch to heat shock treatment at 1, 1.5 or 2 dpf, and periodically checking their survival over time (Fig 7G-I). As expected, the different treatments showed a very mild effect on the WT embryo survival, with a higher mortality rate observed in those embryos treated at the earliest stages of development. Conversely, the TKO embryos presented a higher sensitivity to the treatments in all cases, and a dramatic increase in their mortality rate compared to control embryos. Treatment induced a quick and almost complete lethality of the whole TKO population at later stages, suggesting a very strong susceptibility to stress (Fig 7E-I).

Overall, these data indicate that the lack of *tfeb, tfe3a* and *tfe3b* transcription factors sensitize the embryos to different stress conditions *in vivo* and suggest a potential effect on multiple organs and embryonic tissues.

## Discussion

The present study shows that simultaneous ablation of zebrafish *tfeb*, *tfe3a* and *tfe3b* transcripts induces tissue-specific phenotypes and highlights the pivotal role of *tfe3a* maternal long forms for embryo survival and development. By using a combination of proteomic, scRNAseq transcriptomic, confocal, and electron microscopy analysis, we characterized cellular alterations in the pancreas, liver and gut of knockout embryos and adult animals. The more dramatic defects were found in pancreas, where the accumulation of abnormal zymogen granules led to acinar atrophy in embryos and pancreatitis-like phenotypes in adults. Finally, we showed that the lack of *tfeb* and both *tfe3a* and *tfe3b* transcripts increased the susceptibility of the embryos to different stress conditions, including oxidative stress and prolonged heat shock treatment.

The high level of redundancy between TFEB and TFE3, together with the early lethality of the TFEB knockout embryos in mice, has made nearly impossible assessing the contribution of these transcription factors during embryonic

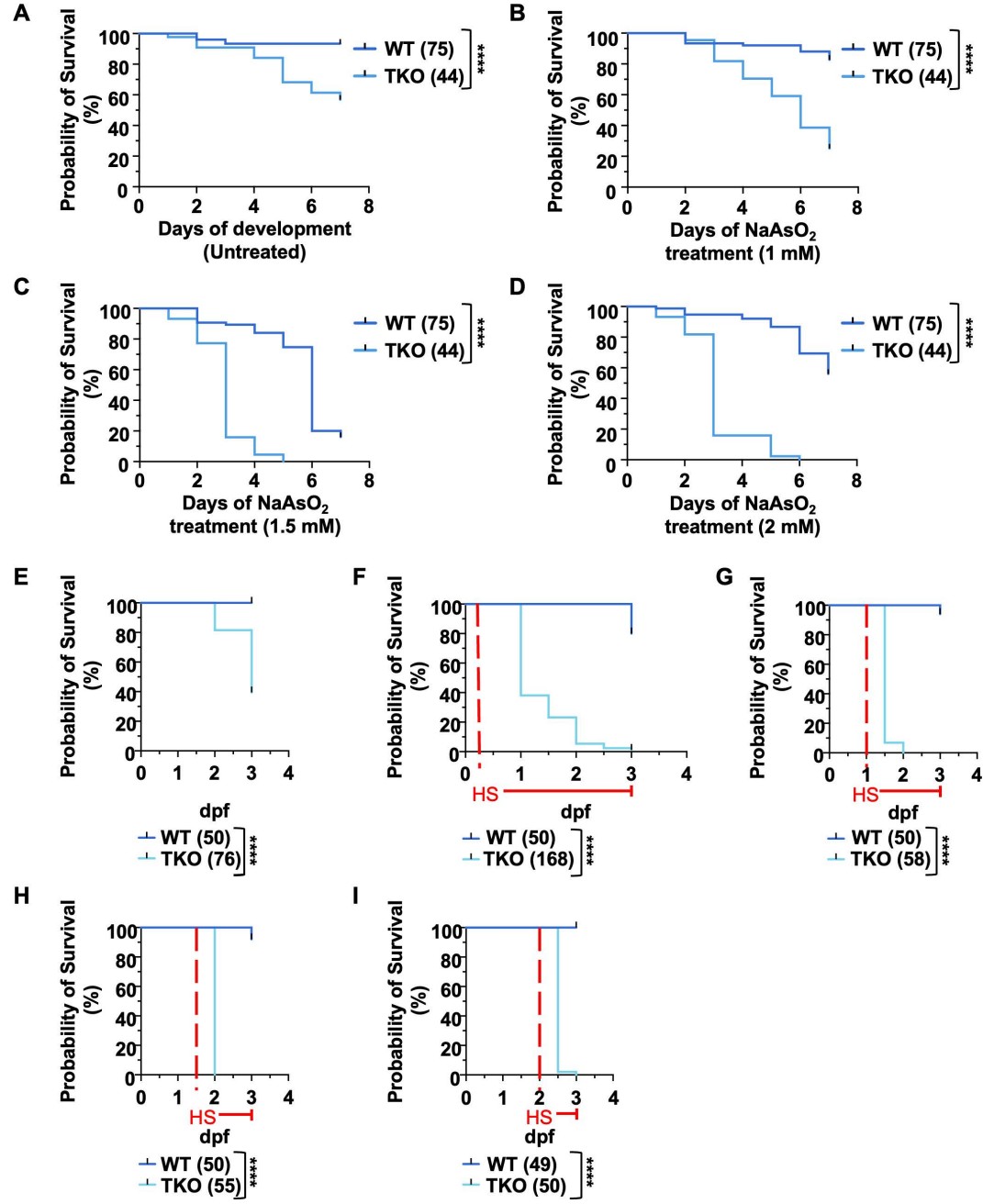

**Fig 7. TKO embryos show higher sensitivity to different kinds of cellular stress. (A-D)** Kaplan-Meier curves comparing the survival rates of WT and TKO embryos at increasing doses of NaAsO$_2$ (untreated (A), 1mM (B), 1.5 mM (C), 2 mM (D)). At each dosage, WT embryos were used as controls. The number of embryos per group is indicated in parenthesis. **(E-I)** Kaplan-Meier curves showing the survival rates of WT and TKO embryos following continuous exposure to heat shock treatments until 3 dpf. The red dashed lines highlight the beginning of each heat-shock (HS) treatment. Untreated embryos (E) were used as a control for the quality of the batch of embryos used for the treatments. In each treatment, WT embryos were used as internal controls. The number in parenthesis indicate the embryos used in each treatment. Logrank tests: ****<0.0001. Data show one representative experiment out of three independently performed.

development. For this reason, we used a highly optimized high-throughput mutagenic strategy to induce genome editing using the CRISPR-Cas9 system targeting all the *tfeb* and *tfe3* long and short isoforms. We found that *tfeb, tfe3a* and *tfe3b* are essential for embryo survival during the early stages of development, with an almost complete lethality of the knockout larvae by 8–10 dpf. Notably, the presence of the *tfe3a* maternal long form, alone, was sufficient to restore embryo survival, allowing them to reach adulthood, even though the animals exhibited reduced growth and severe infertility due to under-developed oocytes. In contrast, the presence of all *tfe3a* forms (maternal and zygotic, long and short) was sufficient to guarantee the embryos a correct development and growth, thus revealing a fundamental role of *tfe3a* in zebrafish survival and development.

We also observed gender-specific phenotypes affecting the growth of male TKO adult fish. The progressive recovery of size specifically observed in TKO female fish is intriguing and seems to suggest the involvement of sexual-related factors. Long-range regulation of the regenerative capacity of organs and appendages has been observed in different species, from low vertebrates to some mammals [55]. Sexual hormones are involved in modulating tissue regeneration in zebrafish, including fin and cardiac regeneration [56,57]. Nevertheless, since different factors might be also involved at local level [58], further studies are required to elucidate the molecular mechanisms at the base of the size recovery observed in female TKO fish.

A previous report showed that *tfeb* and *tfe3* zebrafish genes were necessary for the activation of lysosomal pathways during stress conditions; however, genetic ablation of the long *tfeb* and *tfe3* forms did not affect embryonic survival and development [59]. This suggests that the *tfeb* and *tfe3a* short isoforms may have a compensatory effect when the long forms are absent, which may seem surprising, considering that the short forms lack the transactivation domain. Nonetheless, a previous report showed that a murine version of the TFE3 short form, which is generated by alternative splicing and lacks the transactivation domain, retains some transcriptional activity [60]. Alternatively, the short isoforms may heterodimerize with other members of the MiT-TFE family of transcription factors [61,62], promoting transcription of target genes.

TKO embryos present increased levels of apoptotic cells in basal conditions, mostly in brain and retina, and severe defects to gastro-intestinal organs, in particular exocrine pancreas and liver, which overall may be responsible for their embryonic lethality. Notably, we did not observe major signs of apoptosis in gastro-intestinal organs during development, suggesting that some of the effects caused by the lack of these transcription factors are cell-context dependent, as previously observed in other systems [1]. Proteomic analysis showed a reduction of pancreatic digestive enzymes and lysosomal proteins in TKO embryos. Further histological and EM analysis highlighted the presence of damaged pancreatic zymogen granules in both embryonic and adult TKO samples, as well as typical signs of acute pancreatitis pathogenesis, such as ER stress, activation of the unfolded protein response (UPR), damaged zymogen granules, and accumulation of undigested granules in autophagolysosomes [63].

Pancreatic acinar cells must maintain elevated levels of protein synthesis and secretion, thus requiring a high degree of coordination between different organelles. Particularly relevant is the role played by the autophagic/lysosomal pathway in maintaining acinar cell homeostasis. Pancreatic enzymes or zymogens activate only after they reach the duodenum. Rupture of zymogen granules may lead to premature activation of the enzymes within the pancreatic cells, causing gland autodigestion and pancreatitis. It is therefore essential that defective granules are efficiently removed by autophagy. Consistently, it has been reported that disruption of autophagy by depletion of ATG5 or ATG7, as well as impairment of lysosomal function, is sufficient to cause spontaneous pancreatitis in mice [64–66]. The significant accumulation of abnormal granules in TKO acinar cells, many of them presenting disrupted membranes, is consistent with defects in selective autophagy of zymogen granules or zymophagy. Importantly, *sqstm* and *vmp1*, two of the genes showing a stronger downregulation in TKO acinar cells, have been directly implicated in the removal of fragile zymogen granules [38]. Reduced levels of other important autophagic (*optn*, *ulk1a*, *ulk1b*, *ubc*, *clec16a*) and lysosomal (*cd63*, *ctsla*, *ctsd*, *chmp4ba*) modulators may further contribute to the acinar cell defects observed in TKO animals.

Autophagy is also important to prevent ER stress in acinar cells [67] and loss of pancreatic VMP1 causes ER stress and severe pancreatitis in mice [43,68]. The increased expression of the main regulators of the three arms of the UPR (*xbp1*, *atf6*, and *eif2ak3*) observed in TKO acinar cells, suggests alterations in ER homeostasis. However, the levels of *atf4a* and *atf4b*, as well as many of their targets, were reduced in TKO animals. This is consistent with previous work from our laboratory identifying atf4 as a direct TFE3 target and showing that TFE3 is required to induce transcriptional upregulation of ATF4 in response to ER stress and starvation [44]. Furthermore, mice carrying knockout mutations in *Atf4* showed defects in the development of the exocrine pancreas, including a substantial decrease in the number and size of zymogen granules in acinar cells [69], which is again in agreement with the reduction of the zymogen granules' diameter seen in TKOs.

Consistent with our observations, two recent reports have shown that acinar cell-specific TFEB-KO mice are more susceptible to stresses that induce pancreatitis, such as cerulein and alcohol consumption [39,40]. Our results expand these observations by revealing an essential role of *tfeb* and *tfe3* in providing basal quality control to prevent acinar cell damage during early stages of embryonic development. In addition, it is important to note that whereas different animal models have been generated to study the physiopathology of acute and chronic pancreatitis, most of them involve the use of rodents, and only a few have been developed in zebrafish [70,71]. From that point of view, our mutants might represent a new tool to study pancreatitis pathobiology *in vivo*, particularly given the scarcity of zebrafish genetic models available [41].

Compared to pancreatic acinar cells, liver and gut organs presented cell-cycle alterations in some cellular subpopulations. A previous study in mice showed that the specific knock-out of *TFEB* in endothelial cells (EC) reduced their proliferation *in vivo*, and blocked transition from G to S phase in cultured human primary ECs *in vitro* [17]. These data seem to fit with our scRNAseq analysis of liver dataset at 4 dpf, where we observed a higher percentage of hepatocytes in the $G_1$ stage and a reduction in other stages (S, $G_2$ and M). However, although the *in vivo* confocal analysis using a dual Fucci transgenic line partially confirmed the reduced proliferation in TKO embryos, it also suggested a possible block in the transition from $G_2$ to M phases, at least for a subpopulation of hepatic and gut cells. Future analyses will be focus on clarifying these effects are cell-context dependent or represent a more general condition, as well as the relative involvement of these genes.

Our proteomic and scRNAseq analysis also revealed metabolic reprogramming and increased oxidative stress in both, embryonic and adult TKO hepatocytes. Embryonic hepatocytes seem to be the most affected, which is coherent with the severity of the phenotypes observed at the subcellular level. Although alterations were less pronounced in adult tissues, we did find evidence of increased oxidative stress, such as an upregulation of proteins involved in GSH metabolism. In addition, the second most upregulated protein in the adult liver proteomic dataset was glucose-6-phosphate dehydrogenase (G6pd), the rate-controlling enzyme of the pentose phosphate pathway [48], which plays an important role in sustaining the cellular redox state [72] and gets upregulated in response to oxidative stress [73].

Several studies have reported a role of TFEB and/or TFE3 in the regulation of metabolic functions in different cellular contexts [74,75]. The analysis of tfeb/tfe3 double knockouts in mice showed that they cooperate in controlling the energy metabolism through the regulation of glucose homeostasis, lipid metabolism, and mitochondrial dynamics [25].

Finally, we showed that the lack of *tfeb*, *tfe3a* and *tfe3b* genes strongly increased the sensitivity of the embryos to different environmental stimuli, such as increased oxidative stress and heat shock treatment, further corroborating the pivotal role of these transcription factors in cellular stress responses and animal survival.

## Materials and methods

### Ethical statement

All zebrafish experiments were performed in compliance with the National Institutes of Health (NIH) guidelines for animal handling and research under NHLBI Animal Care and Use Committee (ACUC) approved protocol H-0252R5.

## Zebrafish lines and maintenance

Zebrafish handling and breeding were performed in accordance with relevant guidelines and regulations. All embryos were raised in an incubator at 28.5 °C until 6–7 dpf. Larvae and adult zebrafish were housed in a recirculating aquatic system with a light/dark cycle (14/10 h, respectively) and a water temperature of 28 °C. Zebrafish embryo staging was performed as described previously [76]. The following zebrafish strains were used: wild-type zebrafish strain EK (Ekkwill) and TAB5; *2CLIP;Tg(ptf1a:EGFP)*[jh1] [37] and Tg(-3.5ubb:Cerulean-gmnn-2A-mCherry-cdt1) dual Fucci [45] transgenic backgrounds. During development the living embryos were observed with a Zeiss Discovery.V12 stereomicroscope (Zeiss) with a PlanApo S 0.63X lens (Zeiss). Pictures were taken with an AxioCam ERc5s in Zen 2.3 Pro software (Zeiss).

## Generation of zebrafish *tfeb, tfe3a* and *tfe3b* KO lines

*tfeb* and *tfe3a/b* knockout (TKO) mutants were generated using high-throughput CRISPR/Cas9 methods as previously described [12,77,78]. Briefly, multiple gRNAs and the CAS9 were directly coinjected into the original *tfeb* Ex1 mutant line, and multiple mutated heterozygous alleles for each new site were recovered. To confirm the mutations at genomic level we used directly Sanger sequencing of fluorescent-PCR fragments amplified from gDNA obtained from fin-clipping of the caudal tail of the adult F0 fish. Because heterozygous samples present a mix of WT and mutated alleles which correspond to double peaks starting from the mutation site, the sequencing results were manually annotated using the zebrafish genomic reference present in Ensembl database. The sequencing of mRNA transcripts has been performed using cDNA synthetized from total RNA extracted from whole embryos obtained from an intercross of triple heterozygous F1 fish. Fluorescent PCR products were amplified using the corresponding cDNAs and the fragments were purified, directly Sanger sequenced and manually annotated, if necessary. Triple-heterozygous *tfeb/tfe3a/tfe3b* fish were intercrossed and different combination of genotypes were selected by genotyping. Finally, TKO were intercrossed to obtain lack of maternal transcripts during embryonic development. For a list of the targets used see S8 Table.

## Total RNA extraction from zebrafish embryos and cDNA synthesis

Zebrafish embryos at each stage were euthanized and then homogenized using a 1 ml syringe with a 25 G needle followed by further homogenization using the QIAshredder homogenizers (Qiagen). Total RNA was extracted using the PureLink RNA Mini Kit (Invitrogen) and digested with DNase I and the cDNA synthesis was carried out with the Superscript III First-strand Synthesis SuperMix for qRT-PCR (Invitrogen) using 1 µg of total RNA from each sample following standard conditions. All the primer pairs were designed on different exons to avoid the amplification of DNA possibly contaminating cDNA preparations. RT-PCR products were then separated on agarose gels at various concentrations (from 1 to 3% maximum, based on the fragments length) and visualized by ethidium bromide staining. Lack of gDNA contamination in cDNA samples was confirmed using a fragment of zebrafish β-actin cDNA that was amplified by PCR (35 cycles) as previously described [79]. For the list of the primers see S8 Table.

## Quantitative real-time PCR

Quantitative real-time PCR analyses were performed in quintuplicate using SYBR Green PCR Master Mix (Applied Biosystems) and primers designed using Primer3web (version 4.1.0). Reactions were assembled in 384-well plates (Applied Biosystems) and run under standard conditions on a QuantStudio 12K Flex Real-Time PCR System (Applied Biosystems). Each experiment was replicated at least three times. Gel electrophoresis was used to check the specificity of the PCR products. A single product with the expected length was detected for each reaction. Expression levels were displayed relative to control conditions and normalized using elongation factor 1 alpha (*ef1a*) housekeeping gene expression and using the ΔΔCT method. See S8 Table for the primer sequences.

## Whole mount in situ hybridization (WISH)

WISH and imaging WISH were carried out as described by Thisse [80]. The following DIG-labeled antisense mRNA probes were generated by using DIG RNA Labeling kit (Roche): *fabp10a*, *trypsin* (*try*), *preproinsulin* (*ins*), *tfe3a* (all iso-forms). *tfe3a* WISH antisense probe was generated amplifying a portion of *tfe3a* gene directly from genomic DNA, while the *fabp10a*, *trypsin* and *insulin* antisense probes were created amplifying a portion of the respective cDNAs and cloning them in PCR4-TOPO TA cloning vector (Thermo Fisher Scientific). All embryos used for WISH were fixed overnight in 4% paraformaldehyde (PFA)/PBS at 4 °C, rinsed with PBS-Tween, dehydrated in 100% methanol, and stored at −20 °C until being processed for WISH. Hybridized probes were then detected by using an anti-DIG antibody conjugated to alkaline phosphatase (AP; Roche) at a 1:5,000 dilution. Nitro blue tetrazolium/5-bromo-4-chloro-3-indolyl phosphate (NBT/BCIP; Roche) or INT/BCIP Stock Solution (Roche) was used as a substrate for alkaline phosphatase. Stained embryos were stored in 4% PFA until imaging. Imaging and embryo observation was done using a Zeiss Discovery.v12 stereomicroscope (Carl Zeiss Inc., Thornwood, NY, USA) or a Leica M205 microscope, Leica DFC7000G camera, and the Leica Application Suite X Imaging Software Suite version 3.4.1 (Leica Microsystems, Deerfield, IL, USA). Hybridization was performed at 68 or 70 °C.

## Glycogen quantification

FIJI [81] imaging software version 2.1.3 was used to quantify the total area of glycogen lacune in hepatocytes in electron microscopy pictures. Total glycogen amount of 4 and 5 dpf zebrafish embryos was measured as the amount of glucose released after glycogen digestion with Aspergillus niger amyloglucosidase (MilliporeSigma) as previously described [82,83].

## TUNEL assay

A TUNEL assay to label apoptotic cells was performed using the ApopTag Fluorescein In Situ Apoptosis Detection Kit (Millipore-Sigma, S7110) following the manufacturer's instructions with minor modifications [79] and then imaged using a Zeiss Discovery.v12 stereomicroscope (Carl Zeiss Inc., Thornwood, NY, USA) or Zeiss 880 microscope (Jena, Germany). For the confocal imaging of TUNEL samples, either a Zeiss 20x Plan-Apochromat (0.8 NA) or a 10x Ec Plan-Neofluar (0.3 NA) objective were used. Excitation in the blue channel was performed at 405 nm with power settings below 6.6%, and emission was collected within a bandwidth of 415–475 nm. For the green channel, excitation was conducted at 488 nm with power settings below 2.5%, and emission was collected within a bandwidth of 499–552 nm. Images were captured in a 1024 x 1024 pixel format, with pixel sizes of 0.52 microns in the XY dimension for the 20x objective and 0.83 microns for the 10x objective. Z-stacks were acquired with either 3.0 micron step sizes for imaging with the 20x lens or 4.93 micron step sizes for imaging with the 10x lens. Data acquisition was performed sequentially with a line average of 4.

## Histology

Paraffin sectioning and hemotoxin and eosin staining of adult fish were performed by Histoserv, Inc. (Germantown, MD). For paraffin consecutive sections of zebrafish embryos, the embryos at specific stages were fixed in 4% paraformaldehyde/PBS overnight at 4 °C. After 5 washes 1x in PBS, the embryos have been dehydrated with a scale of ethanol (25%, 50%, 75%, 95% and 100%) and then clearified in a scale of Xilene (ethanol:xylene 3:1; ethanol:xylene 1:1; ethanol:xylene 1:3; xylene 100%; xylene 100%) under a chemical hood. Cleared larvae were infiltrated and embedded in paraffin Paraplast X-tra (Electron Microscopy Sciences) in an oven at 80°C. Embryos were moved to paraffin blocks (Electron Microscopy Sciences) and manually oriented before the solidification of the paraffin. Consecutive five μm thick sections were taken on Superfrost coated slides (Fisherbrand) using a microtome. Paraffin on the slides was removed by passing them through three xylenes washes. After deparaffinization, the tissues were re-hydrated through a decreasing series of

alcohol (100%, 90%, 75%, 50%, 25%) then moved to distilled water and stained with hemotoxin and eosin and mounted. Images of all the slides were acquired using a Hamamatsu NanoZoomer 2.0-RS slide scanner using a 20x variable lens (Hamamatsu).

## Proteomics

Comparative proteomic analysis of WT and TKO embryos at 5 dpf, as well as liver explanted from WT and TKO ~ 1 year old adult male fish, was performed using the TMT-18plex labeling based quantitative proteomics method and liquid chromatography-tandem mass spectrometry (LC-MS/MS) by Poochon Scientific (Frederick, MD). Briefly, frozen samples were processed and, after protein quantification, 100 µg of protein from each sample was processed for trypsin digestion, followed by TMT-multiplex labeling using one TMT-18plex set. Seven unique TMT tags were used to label 50 µg of trypsin-digested peptides from each of the 6 digests/experiment and one master mix. After TMT labeling, the 7 labeled peptides were mixed followed by fractionation using basic reversed-phase UHPLC. Twenty-four fractions from each set were generated and analyzed sequentially by 24 110-minute LC-MS/MS runs. The 24 raw MS data files acquired from analysis of the 24 fractions were searched against Zebrafish (*Danio rerio*) protein sequences database obtained from the Uni-protKB website using Proteome Discoverer 2.5 software (Thermo, San Jose, CA) based on the SEQUEST and percolator algorithms. The false positive discovery rate (FDR) was set to 1%. The resulting Proteome Discoverer Report contains all assembled proteins with peptides sequences and peptide spectrum match counts (PSM#), and TMT-tag based quantification ratio. Through comparison of the abundance of the 7860 proteins between 2 groups (WT and TKO), the proteins that significantly changed were analyzed and ranked by a fold change ≥25% and $p < 0.05$ (n = 3). Student's t-test was used to analyze the statistical significance of differences between each two groups (WT and TKO) for comparison. The Average, STDEV, CV, and paired t-test (p-values) were calculated using Microsoft Excel. Zebrafish (*Danio rerio*) protein sequences Database was downloaded from the UniprotKB website.

## Gene ontology (GO) analysis

GO Enrichment analysis was performed using ShinyGO 0.80 (http://bioinformatics.sdstate.edu/go/) [84] and Enrichr (https://maayanlab.cloud/Enrichr/) [85].

## Two-photon zebrafish imaging

All imaging was conducted using a Leica SP8 Two Photon DIVE upright microscope (Leica Microsystems, Mannheim, Germany) with a pulsed dual-beam Insight X3 Ti-Sapphire laser (MKS Spectra-Physics, Milpitas, CA) and a Leica 25x HC FLUOTAR water dipping lens (NA 1.0). The pinhole diameter was set to 1 Airy Unit (A.U.), and the image format was set to 1024 x 1024 pixels. GFP imaging was performed using a two-photon excitation wavelength set to 920 nm at 0.6% laser power (< 1 mW at the back aperture), with fluorescence emission collected within a 500–550 nm bandwidth using a Leica HyD detector in the non-descanned emission pathway, set to a gain of 25%. The pixel size was 0.19 microns in the XY dimension. Bidirectional scanning was employed at 600 Hz. Tiled image volumes were collected with individual tiles covering 200 microns in the XY dimension and 1.0 microns in the Z dimension, with an imaging depth of approximately 150 microns.

The zebrafish FUCCI line was imaged with 870 nm excitation for the CFP component at 6.5% laser power (approximately 28 mW at the back aperture) and 7.3% laser power for the WT and KO lines, respectively, with 50% gain on the HyD detectors. For the mCherry component, 1045 nm excitation was used at 5% laser power (approximately 120 mW) and 2.7% laser power (approximately 70 mW) for the WT and KO lines, respectively, with 130% gain on the HyD detector. A line scan average of two was set to reduce noise in the image. For confocal movies, time intervals of 10 min 7 sec for the WT and 7 min 44 sec for the TKO were used.

All images were collected using Leica Application Suite X (LAS X) software. Volumetric reconstructions were made using the Imaris software package version 10.1.0 (Oxford Instruments, Abingdon, UK).

## Transmission electron microscopy

For electron microscopy analysis, 5 dpf zebrafish embryos and liver and pancreas organs explanted from 7 months old male fish were fixed at RT for 1 hour then overnight at 4 °C in 2.5% Glutaraldehyde + 4% Paraformaldehyde in 0.1M Sodium-Cacodylate buffer, pH 7.4. After fixation, samples were washed in 0.1M Sodium-Cacodylate buffer 3x20 minutes and postfixed in 1% $OsO_4$ and 1.5% $K_3Fe(CN)_6$ in 0.1M Sodium-Cacodylate buffer, pH 7.4 for 1 hour on ice. Samples were then rinsed and washed 2x10 minutes with water and kept overnight in 1% Aqueous UA at 4 °C. The following day, samples were rinsed in water once and washed for 2x10 minutes, dehydrated in a graded series of ethanol for 10 minutes each (30%, 50%, 70%, 90% and 100% twice) followed by propylene oxide and infiltrated with Epon resin (Embed 812, Electron Microscopy Sciences) using increasing concentrations of resin dissolved in propylene oxide and three changes of 100% resin for 2 hours each. Solidification of the resin was obtained baking at 60°C for 48 h. Semithin sections (1–2 mm) were initially obtained and stained with toluidine-blue to isolate the region of interest. Ultrathin sections (65 nm) were cut from the block surface using a Leica EM UC7 ultramicrotome and digital micrographs were acquired on a JEOL JEM 1200 EXII TEM operating at 80 kV and equipped with an AMT XR-60 digital camera.

## FACS-sorting

Wild-type and TKO embryos were collected at 4 dpf, and GFP+ or dsRED+ cells from dissociated embryos were sorted on a FACS AriaIII (Becton Dickinson, Franklin Lakes, NJ) using FACS DIVA software (Version 8.0.20). For GFP+ cells, a manual resection of the tail and the head of each embryo was performed to enrich the pancreatic population of GFP+ cells. Single cell suspensions were generated using a previously published protocol [86] and stained with eBioscience Fixable Viability Dye eFluor 780 (Thermo Fisher Scientific) before the sorting following manufacter's instruction to label dead cells and then sorted. Sorted cells were first gated on Forward Scatter Area versus Side Scatter Area dot plot. Two additional dot plots were used to remove cell clumps by using a singlet gate on Forward Scatter Area versus Forward Scatter Width followed by a second singlet gate on Side Scatter Area versus Side Scatter Width. Cells were also gated to select live that were negative fluorescence for the Viability Dye eFluor 780 (633nm excitation, emission 780/60nm). Cells falling within these four gate regions were then evaluated for GFP+ cells on a dot plot of GFP (488nm laser excitation, emission 530/30nm bandpass) versus autofluorescence (488nm laser excitation, emission 576/26nm bandpass). The dsRED+ cells were identified on a plot of dsRed (561nm laser excitation, emission 610/20nm bandpass) versus GFP (488nm laser excitation, emission 530/30nm bandpass). GFP+ and dsRED+cells were collected separately in a small volume of DMEM-10%FBS (25–30 ul) in a PCR-tube at room temperature and the final volume was then adjusted according to the 10X Chromium protocol (see below).

## Single cell capture, and sequencing with 10X genomics chromium

Single cell suspensions quality, number and viability were assessed with a dual fluorescence cell counter LUNA-FX7 (Logos Biosystems) using Trypan Blue Stain, 0.4% (cat T13001, Logos biosystems). 10,000 cells were targeted from each sample cell suspension. Cells were washed twice with PBS + 0.04% BSA and resuspended in about 500 cells per microliter. 10X Genomics's Chromium instrument and Dual index Single-Cell 3′ Reagent kit (V3.1) were used to prepare the individually barcoded single-cell RNA-Seq libraries following the manufacturer's protocol. Libraries quality was assessed by the TapeStation-4200 traces (Agilent BioAnalyzer High Sensitivity Kit) and quantitated by the Qubit system. Sequencing was done on the Illumina NextSeq-550 machine, using the High Output Kit v2.5 (Illumina). Following sequencing, an average 25,000 reads per cell were generated. The bcl files were demultiplexed into a FASTQ, aligned to Danio.rerio_genome and single-cell 3′ gene counting were performed by the standard 10X Genomics's CellRanger mkfastq software (V7.0.1). The single-cell QC was generated by 10X Genomics's CellRanger and visualized using 10X Genomics's Loupe Cell Browser. Valid barcode reads in all samples were >95%. Reads aligned to zebrafish genome were >70% in all samples,

while anti-sense reads were <2% in all samples. All key metrics were within the expected range and did not trigger any errors or warnings in the Cell Ranger web summaries.

## scRNAseq data processing and clustering

Clustering, filtering, variable gene selection, and dimensionality reduction were performed using the Seurat v3.2 [87,88]. Raw data matrices of 2 or more Seurat objects were merged to generate a new Seurat object with the resulting combined raw data matrix. The cell identities of the clusters were reassigned based on the expression of well-established hematopoietic markers that were identified by our unbiased analysis as cluster signature genes. scRNA-seq data have been deposited in GEO (GSE278733).

For all our analysis raw data matrices of two or more Seurat objects were merged to generate a new Seurat object with the resulting combined raw.data matrix. For the GFP+ cells, the original identifiers for each dataset were set and only recovered at the end of the analysis. Cells with <200 and >4000 detected genes or with >5 percentage of UMIs mapped to mitochondrial genes were excluded from further analysis. For the dsRED+ cells, the original identifiers for each dataset were set and only recovered at the end of the analysis. When we checked the mitochondrial QC metrics, both WT and TKO sorted cells presented a percent of mitochondrial DNA higher than usual. Because hepatocytes may contain high number of mitochondria [49], we set a < 60% of percent.mt as a threshold to include the hepatocytes in the analysis. The Seurat objects were processed according to the Seurat-guided clustering tutorial at https://satijalab.org/seurat/vignettes.html. Differential expression analysis between the WT and TKO GFP+ or dsRED+ cells was done using Seurat default function "FindMarkers". scRNA-seq data are available at GEO under accession number # GSE278733.

## NaAsO$_2$ and heat-shock treatments and survival assay in zebrafish embryos

For NaAsO$_2$ treatment, WT and TKO fish were crossed and at 24 hpf, embryos were moved to E3 medium and divided in different 60 mm tissue culture dishes (Falcon) based on the size of each clutch. The medium was removed and changed with different concentrations (1, 1.5 and 2 mM) of a NaAsO$_2$ solution (ChemCruz, CAS number 7784-46-5) in E3 medium from a 1 mM stock solution in ultra-pure water. Embryo survival was checked every morning under a stereomicroscope (Zeiss), dead embryos were removed, and the solutions were changed with new ones prepared fresh every day from a 1 M NaAsO$_2$ stock solution prepared daily diluting the powder in ultra-pure water. At the end of each treatment the survival embryos were anesthetized and euthanized. For heat-shock treatments, WT and TKO fish were crossed and the day after, embryos were collected, moved to E3 medium, divided in different 100 mm tissue culture dishes (Falcon) based on the size of each clutch, and kept at 28.5°C until further processing. At specific time points (5 hpf, 1, 1.5 and 2 dpf), part of each clutch was moved to a different temperature setting (33°C for 5 hpf and 37°C for all the rest) in a specific incubator. Embryo survival was checked every ~12 hours under a stereomicroscope (Zeiss), dead embryos were counted and removed. Fresh pre-warmed E3 medium was added each day. At the end of each treatment (3 dpf) all embryos were anesthetized and euthanized. For both, NaAsO$_2$ and heat-shock treatments, collected data was used to calculate survival rates using the Kaplan-Meier method in Prism 10 (Graphpad). Statistical analysis was performed on pairwise comparison of individual survival curves using the logrank (Mantel-Cox) test.

## Statistical analysis and reproducibility

Each experiment was performed at least three times independently with similar results unless otherwise mentioned in figure legends. Obtained data were processed in Excel (Microsoft Corporation) and Prism 10 (GraphPad Software) to generate bar charts and perform statistical analyses. Student's *t* test, logrank (Mantel-Cox) test, one-way ANOVA or two-way ANOVA were run for each dependent variable, as specified in each figure legend. All data are presented as mean ± standard error of the mean (SEM), unless otherwise specified. No statistical method was used to predetermine the

sample size. No data were excluded from the analyses. The experiments were not randomized. The investigators were not blinded to allocation during experiments and outcome assessment.

## Supporting information

**S1 Fig. Lack of *tfeb* and *tfeb*, *tfe3a and tfe3b* induced altered gene expression and apoptosis during embryo development. (A)** Mutated alleles characterizing the genotype of TKO fish. **(B)** Genomic and mRNA regions around each gRNA target were amplified by PCR and then SANGER sequenced. Part of the sequenced regions surrounding each site has been compared to the WT sequences and is shown in each alignment. Lower case letters indicate intronic sequences, bold letters highlight single base insertions. Asterisks and dashes mark identical and deleted bases, respectively. **(C)** Real-time qPCR expression analysis of different *tfe3a* maternal forms at 8/16 cell stage in WT and embryos obtained from different crosses. The expression in WT embryos has been used as reference. **(D)** Relative expression analysis by real-time qPCR of long and short *tfeb* maternal forms at 8 cell stage in WT and embryos obtained from a cross of females TKO and males DKO-*tfe3a*$^{+/-}$. **(E-F)** Analysis of lysosomal markers in TKO and WT embryos at 2 and 4 dpf. **(G)** Proliferation, cell cycle, pro- and anti-apoptotic expression markers at 4 dpf. Proliferation marker: *pcna*. Cell proliferation inhibiting factors: *ccng1*, *p21*, *Delta113tp53*. Pro-apoptotic genes: *tp53*, *rps27l*, *mdm2* and *baxa*. Anti-apoptotic genes: *bcl2l1*. All the data represent means ± SEM, n = 3 independent experiments. Statistical significance was determined by using two-way ANOVA with Sidak's multiple comparisons. ns, not significant, * < 0.05, ** < 0.01, *** < 0.001, **** < 0.0001. **(H-I)** TUNEL assay in WT and TKO embryos at different developmental stages. (H) Confocal maximum projections of embryos showing TUNEL$^+$ cells (green) and DAPI (blue) at 3 and 4 dpf. Lateral positions, anterior to the left. Scale bars, 100 μm. (I) Fluorescent stereomicroscope pictures of TUNEL staining (green) of 5 dpf embryos in lateral and dorsal orientation, anterior to the left. Scale bars, 150 μm. E, eye; OT, optic tectum; PF, pectoral fins.
(TIFF)

**S2 Fig. Lack of *tfeb*, *tfe3a and tfe3b* affects pancreas and liver development. (A)** Heat map showing the relative abundance of 856 differentially expressed ranked proteins (fold change >1.5, p < 0.05, n = 3). The color key indicates the relative abundance of each protein (0 to 1.0) across 6 samples. **(B-C)** ShinyGO KEGG (B) and Molecular Function (C) term enrichment of proteins significantly upregulated in TKO mutants compared to WT siblings. FDR, false discovery rate. **(D)** Heat map showing the relative abundance of vitellogenin proteins in WT and TKO samples. **(E)** H&E staining of longitudinal sections showing the pancreas and gut in the trunk of paraffin embedded WT and TKO embryos at 4 dpf. G, gut; I, islet; P, exocrine pancreas. Scale bars, 50 μm. **(F-I)** H&E staining of transversal sections showing the gastro-intestinal organs in the trunk of paraffin embedded WT (F-G) and TKO (H-I) 5 dpf embryos. Consecutive sections are shown in rostro-caudal order from left to right and from top to bottom of the sequence. Black arrows point to discontinuities in the exocrine pancreas tissue, black arrowheads indicate the central region of exocrine pancreas. P, exocrine pancreas; G, gut; L, liver. 5 μm sections. Scale bars, 100 μm.
(TIFF)

**S3 Fig. Morphological defects TKOs embryonic pancreas. (A-B)** Representative max-projections (green) and Imaris 3D reconstructions (yellow) of confocal Z-stacks of 2CLIP WT and TKO embryos at 4 and 5 dpf, respectively. Dorsal views, anterior to left. Scale bars, 30 μm. **(C)** Dorsal views of 4.5 dpf WT and TKO 2CLIP embryos. Red dashed lines indicate the midline of the embryos, pancreas have been highlighted with a yellow line. The yellow arrow points to GFP$^+$ signal on the other side of the embryo near the liver (white asterisks), suggesting possible heterotaxia. **(D)** Representative max-projections (top) and corresponding Imaris 3D reconstructions (bottom) of confocal Z-stacks of TKO L embryos at 5 dpf. Dorsal views, anterior to the left. **(E)** Quantification of the percentage of embryos for each category in WT and TKO embryos at 4.5 dpf. L, lefty; R, righty; SI, Situs Inversus. **(F-G)** Toluidine staining of resin semithin transversal sections of

5 dpf embryos (F) and a portion of adult pancreas (G) showing the regions used for the TEM analysis. Liver and pancreas are highlighted in yellow and red, respectively. Scale bars, 50 μm.
(TIFF)

**S4 Fig.** Single cell RNA-sequencing data from sorted GFP+ cells at 4 dpf. **(A)** Experimental workflow used for scRNA-seq experiments of 4 dpf embryos. **(B-C)** FACS isolation of the GFP+ cell population from 4 dpf WT and TKO 2CLIP embryos. **(D)** Table showing cell quality control metrics for the GFP+ cells generated by 10X Cell Ranger. **(E)** Violin plots showing quality control features (nFeature_RNA, nCount_RNA and percent.mito) for both, GFP+ WT and TKO data sets. **(F)** UMAPs showing clusterization of WT (left) and TKO (right) GFP+ cells. **(G)** UMAPs showing the projection of the clusters (left) and the two genotypes (right, WT and TKO) merged for the analysis. **(H)** Dot plot showing expression of liver, endocrine pancreas (EnP), exocrine pancreas (ExP) and neural markers in the different cluster population. **(I)** UMAPs showing clusters (left) and genotype (right) of the subset of the pancreas-related clusters. Numbers in parenthesis indicate the total number of cells for each cluster. **(J)** Dot plots showing expression of cell cycle ($G_1$/S and $G_2$/M), acinar/endocrine/ductal (Ac/End/Ductal), pancreatic Notch-responsive cells/centroacinar cells (PNCs/CACs), and epithelial-mesenchymal transition (EMT) markers in merged clusters shown in (I). In each dot plot, the size of the dot encodes the percentage of cells within a class, the color indicates the average level of expression (purple is high, grey is low).
(TIFF)

**S5 Fig. Morphological defects in livers of TKO embryos and adult fish. (A)** Cartoon showing the proliferation rate during liver growth phase in WT embryos. Left lobe livers reach cellular homeostasis ~5–7 dpf. **(B)** (Left) Representative max-projection of confocal Z-stacks of a FUCCI WT embryo at 3 dpf. The dashed yellow line highlights the left lobe liver. Lateral views, anterior to the left and dorsal to the bottom. Scale bar, 20 μm. (Right) Pictures of the same single plane over time from each confocal Z-stack. Arrowheads point at the same cerulean+ cells over time; during cell division, arrowheads present an outline of a different color. Note that cerulean+ nuclei become brighter right before the beginning of mitosis. Numbers indicate the time of each frame, starting from 50 minutes. **(C-D)** Representative EM pictures of hepatocytes from WT and TKO adult fish. Red asterisks label cellular peroxisomes. N, nuclei; g, glycogen lacunae. Scale bar, 2 μm (C) and 1 μm (D). **(E)** Quantification of the number of peroxisomes/area (~150 μm$^2$) in EM sections of WT and TKO adult livers. Data represents means±SEM. Statistical significance was determined by unpaired two-tailed t-tests. * < 0.05. **(F)** (Left) A volcano plot showing 245 differentially expressed proteins between TKO and WT zebrafish (n = 3 each). X-axis represents log2 of fold changes, y-axis represents statistically significant p-value (-log10 of p-value, n = 3). Blue dots represent 7615 proteins fold change <1.5, red dots are 245 differentially expressed proteins with a fold change >1.5, p < 0.05. (Right) Heat map showing the relative abundance of the same 245 differentially ranked proteins. Color Key indicates the relative abundance of each protein (0 to 1.0) across the 6 samples. **(G-H)** ShinyGO KEGG term enrichment of those proteins significantly downregulated (G) and upregulated (H) in TKO mutants compared to WT siblings. FDR, false discovery rate.
(TIFF)

**S6 Fig.** Single cell RNA-sequencing data from sorted dsRED+ cells at 4 dpf. **(A-B)** FACS isolation of the dsRED+ cell population from 4 dpf WT and TKO 2CLIP embryos. **(C)** Table showing cell quality control metrics for the dsRED+ cells generated by 10X Cell Ranger. **(D)** Violin plots showing quality control features (nFeature_RNA, nCount_RNA and percent.mt) for both, dsRED+ WT and TKO data sets. **(E)** UMAPs showing clusterization of WT (left panel) and TKO (right panel) dsRED+ cells. The total number of cells in each cluster in shown in parenthesis. **(F)** Dot plots showing expression of hepatic, endocrine pancreas, and neuronal markers in WT (top panel) and TKO (bottom panel) dsRED+ datasets. **(G)** Violin plots showing the quality percent.mt control feature of each cell cluster for both the dsRED+ WT (left) and TKO (right) data sets as shown in (E). **(H)** Dot plots showing expression of liver, endocrine pancreas, neural, embryonic cholangiocytes (Chol), and proliferation markers in merged clusters shown in Fig 6A. **(I)** Cell cycle analysis using Seurat

CellCycleScoring function in WT and TKO hepatic cells. Statistical significance was determined using Fisher's exact test. *<0.05 and **<0.01. **(J)** Gene ontology enrichment analysis of downregulated (DOWN) and upregulated (UP) genes in TKO and WT liver clusters: clusters 1 vs 0 as shown in Fig 6C. In each dot plot, the size of the dot encodes the percentage of cells within a class, the color indicates the average level of expression (purple is high, grey is low). (TIFF)

**S7 Fig.  TKO show reduced survival under cellular stress conditions. (A-B)** Kaplan-Meier curves showing the survival rates of WT (A) and TKO (B) embryos following continuous exposure to different doses of $NaAsO_2$ from 1 dpf stage. At each dose, untreated embryos were used as controls. The number of embryos per group is indicated in parenthesis. The same number of embryos has been used in each treatment. (Logrank tests: n.s., not significant; **<0.01; ***<0.001, ****<0.0001). Data show one representative experiment out of three independently performed. **(C)** Heat map showing the relative abundance of selected proteins at 5 dpf. The color key indicates the relative abundance of each protein (0.2 to 1.0) across the 6 samples. (TIFF)

**S1 Table.  Summary of mutants and their phenotypes.** Summary of all the different combinations of KO mutants analyzed and the survival observed during development and adulthood. In red are highlighted the combinations that presented embryonic lethality, in bold the ones that presented phenotypes during adulthood. (DOCX)

**S2 Table.  Proteomic analysis of 5 days post fertilization WT and TKO embryos.** In this study, 7652 proteins were quantitatively identified. 856 of 7652 proteins are significantly changed in their abundance (expression) (fold change ≥50%; p<0.05; n=3) in TKO group comparing to WT group. The 856 proteins differentially expressed were ordered by their expression level (Log2-Ratio (TKO vs WT)) from the most downregulated to the most upregulated. (XLSX)

**S3 Table.  GO analysis of down- and up-regulated in 5 days post fertilization WT and TKO embryos.** List of differentially expressed proteins in WT and TKO embryos at 5 dpf and their respective GO analysis as shown in the study. Downregulated (DOWN) and upregulated (UP) proteins have been separated in different tables and listed by the most down- and up-regulated, respectively. (XLSX)

**S4 Table.  scRNAseq data on GFP+ cells sorted from 4 dpf WT and TKO embryos.** List of marker genes for each cluster shown in the different UMAPs of GFP+ cells. The last table lists the genes differentially expressed between the TKO and the WT acinar cells (from the most down- to the most up-regulated). (XLSX)

**S5 Table.  Proteomic analysis of WT and TKO adult livers.** In this study, 7860 proteins were quantitatively identified. 245 of 7860 proteins are significantly changed in their abundance (expression) (fold change ≥50%; p<0.05; n=3) in TKO group comparing to WT group. The 245 proteins differentially expressed were ordered by their expression level (Log2-Ratio (TKO vs WT)) from the most downregulated to the most upregulated. (XLSX)

**S6 Table.  GO analysis of down- and up-regulated in 5 days post fertilization WT and TKO embryos.** List of differentially expressed proteins in WT and TKO adult livers and their respective GO analysis as shown in the study. Downregulated (DOWN) and upregulated (UP) proteins have been separated in different tables and listed by the most down- and up-regulated, respectively. (XLSX)

**S7 Table.  scRNAseq data on dsRED+ cells sorted from 4 dpf WT and TKO embryos.** List of marker genes for each cluster shown in the different UMAPs of dsRED+ cells, DE analysis of TKO vs WT hepatocytes and GO analysis of the differentially expressed genes (down- and upregulated, respectively).
(XLSX)

**S8 Table.  List of oligos and gRNAs used in this present study.**
(DOCX)

**S1 Data.  Numerical DATA for figures.**
(XLSX)

**S2 Data.**  Numerical DATA for supplemental figures.
(XLSX)

## Acknowledgments

We would like to thank the staff of Charles River for animal care; Kevin Bishop, Blake Carrington and Stephen Frederickson from the NHGRI Zebrafish Core for the support, Jose Martina for helpful discussions and technical support, and all the members of the Puertollano Lab. We thank Dr. Xufeng Wu for assistance during confocal imaging and Dr. Carmen López-Iglesias and Dr. R.I. Koning for helpful discussions.

## Author contributions

**Conceptualization:** Alberto Rissone, Christian A Combs, Rosa Puertollano.

**Formal analysis:** Vicky Chen, Rosa Puertollano.

**Investigation:** Alberto Rissone, Martina La Spina, Zulfeqhar A Syed, Christian A Combs, Martha Kirby, Abdel Elkahloun, Raman Sood.

**Resources:** Shawn M Burgess, Rosa Puertollano.

**Supervision:** Rosa Puertollano.

**Writing – original draft:** Alberto Rissone, Rosa Puertollano.

**Writing – review & editing:** Alberto Rissone, Martina La Spina, Erica Bresciani, Christian A Combs, Shawn M Burgess, Rosa Puertollano.

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
