## [Decision Letter · Decision Letter 0]

PGENETICS-D-25-00268

The transcription factors Tfeb and Tfe3 are required for survival and embryonic development of pancreas and liver in zebrafish

PLOS Genetics

Dear Dr. Puertollano,

Thank you for submitting your manuscript to PLOS Genetics. After careful consideration, we feel that it has merit but does not fully meet PLOS Genetics's publication criteria as it currently stands. Therefore, we invite you to submit a revised version of the manuscript that addresses the points raised during the review process.

Please submit your revised manuscript within 60 days May 26 2025 11:59PM. If you will need more time than this to complete your revisions, please reply to this message or contact the journal office at plosgenetics@plos.org. Please include the following items when submitting your revised manuscript:

We look forward to receiving your revised manuscript.

Kind regards,

Brian D. Perkins, Ph.D.

Academic Editor

PLOS Genetics

Pablo Wappner

Section Editor

PLOS Genetics

Aimée Dudley

Editor-in-Chief

PLOS Genetics

Anne Goriely

Editor-in-Chief

PLOS Genetics

**Additional Editor Comments:**

All reviewers find the manuscript to be well-written and the study design to be rigorous but some revisions are needed to improve clarity. The authors are strongly encouraged to address all the points raised by the reviewers.

**Journal Requirements:**

https://journals.plos.org/plosgenetics/s/submission-guidelines#loc-parts-of-a-submission

- TM on pages: 27, 33, and 34.

4) Thank you for including an Ethics Statement for your study. Please include:

i) The full name(s) of the Institutional Review Board(s) or Ethics Committee(s).

5) Please upload all main figures as separate Figure files in .tif or .eps format. For more information about how to convert and format your figure files please see our guidelines: 

6) We notice that your supplementary Figures, and Tables are included in the manuscript file. Please remove them and upload them with the file type 'Supporting Information'. Please ensure that each Supporting Information file has a legend listed in the manuscript after the references list.

7) Some material included in your submission may be copyrighted. According to PLOSu2019s copyright policy, authors who use figures or other material (e.g., graphics, clipart, maps) from another author or copyright holder must demonstrate or obtain permission to publish this material under the Creative Commons Attribution 4.0 International (CC BY 4.0) License used by PLOS journals. Please closely review the details of PLOSu2019s copyright requirements here: PLOS Licenses and Copyright. If you need to request permissions from a copyright holder, you may use PLOS's Copyright Content Permission form.

Potential Copyright Issues:

i) Please confirm (a) that you are the photographer of 1C, and 1F, or (b) provide written permission from the photographer to publish the photo(s) under our CC BY 4.0 license.

ii) Figure S4A: Thank you for stating that the figure is created with BioRender. Please confirm that you hold a Premium account and provide a pdf copy of the CC BY 4.0 License as provided by BioRender. For instructions on how to generate a CC BY 4.0 license for your figure, please see the guidelines here: https://help.biorender.com/hc/en-gb/articles/21282341238045-Publishing-in-open-access-resources. 

If you are using the free assets from BioRender, we are unable to publish these images as they are licensed under a stricter license than CC BY 4.0. In this case we ask you to remove the BioRender images and replace them with open source alternatives.

See these open source resources you may use to replace images / clip-art:

- https://bioart.niaid.nih.gov/

- https://bioicons.com/

- https://healthicons.org/

- https://scidraw.io/

- https://reactome.org/icon-lib

- https://www.phylopic.org/images

8) Thank you for stating "The scRNA-seq data have been deposited in NCBI GEO database and assigned the identifier GSE278733." Please note that, though access restrictions are acceptable now, your entire minimal dataset will need to be made freely accessible if your manuscript is accepted for publication. This policy applies to all data except where public deposition would breach compliance with the protocol approved by your research ethics board. If you are unable to adhere to our open data policy, please kindly revise your statement to explain your reasoning and we will seek the editor's input on an exemption.

9) Please amend your detailed Financial Disclosure statement. This is published with the article. It must therefore be completed in full sentences and contain the exact wording you wish to be published.

3) If any authors received a salary from any of your funders, please state which authors and which funders.

**Reviewers' comments:**

Reviewer's Responses to Questions

Reviewer #1: In this study, the authors investigated the roles of the transcription factors TFEB and TFE3 in development by using CRISPR/Cas9-mediated gene deletion of tfeb, tfe3a, and tfe3b in zebrafish. The findings reveal that Tfeb and Tfe3 are essential for maintaining cellular and tissue homeostasis, particularly in the retina, pancreas, and liver during development. The study was well-executed, with high-quality data that support the conclusions. It has the potential for significant impact, as similar investigations have not yet been conducted in other mammalian systems.

Specific comments:

1. On page 23, some references were formatted with author names rather than numbers.

2. TFEB and TFE3 are known to regulate PGC1a expression and mitochondrial biogenesis. It would be beneficial if the authors could provide a more detailed analysis of the PGC1a-mediated pathway within their datasets.

3. Lysosomal functions and biogenesis are closely linked to autophagy. While the authors have included some information on mRNA changes in autophagy-related genes, it would be helpful to show the changes in LC3 and p62 in these embryos.

Reviewer #2: In this manuscript, Rissone et al. describe their generation of zebrafish carrying a triple mutation in the Tfeb, Tfe3a and Tfe3b genes and the resulting phenotype. They show that together, these genes are essential for embryo survival and show defects in brain and retina as well as in pancrease, liver and the gut. The knockout animals accumulate abnormal zymogen granules leading to acinar atrophy and pancreatitis. They also show susceptibility to stress. The manuscript is mostly well written and the results clearly displayed and logical. Figure 1 is rather confusing as the panel on the left only indicates the exons but not the functional domains – which are shown on the right but without reference to the exons. It would be better to indicate the domains and exons in the same figure for all genes. Did the 7bpDEL mutation in the first exon of tfeb affect the transcript in terms of expression levels or alternative products? In the description of the triple mutant they say that they confirm the mutations a the genomic and mRNA level by Sanger sequencing but they do not explain which material they used. Also, they presume that the reader understands that the mutations they generated were homozygous (except in the case of Tfeb exon6 mutation). In the description of the TUNEL staining of embryos the description is not very clear. The authors simply state that “TUNEL+ cells in specific areas of the embryos“ without specifying if they are talking about wt or mutant embryos. On page 11 the authors start using the term tfe3s, presumably for the shorter isoform of tfe3. However, they have not really explained this properly or validated that they are truly dealing with the short isoform. When describing the scRNAseq results they claim that “Analysis of cell genotype in the different clusters...were a mix of WT and TKO cells“. This, however, is not consistent with Figure 4A right panel which shows a clear separation of the WT and TKO cells (dark blue vs purple). Why is this difference? This is acknowledged a few lines futehr down.

Minor comments:

Abstract:

1. “tf3b“ should be “tfe3b“.

2. In the sentence that begins with “These genes were essential...“ it is not clear whether the authors mean these genes individually or together. Needs to be clear.

Introduction:

1. “TFE3-S321 promotes the(n) binding to 14-3-3“

2. A reference is missing for the sentenc “Additional post-translational modifications...“

3. “Compared to mammalIAN spcies, zebrafish....“

4. “factors are essential for larvaL survival“

Results:

1. “...which is the result of two separate(d) events.

2. The 2bpDEL is not shown in Figure S1A or B.

3. The sentence “Since the first attempts to...“ is not very clear. Attempts in previous literature or in this study?

4. Please label the genotypes of the fish in Fig. 1C.

5. In Figure 1E and F the authors talk about length (in cm) and size (in cm). What is the difference between the two variables?

6. It is not clear what the authors mean by “in-cross“

7. “lack of maternal tfe3a transcripts (was) strongly affectED the survival...“ This sentence actually is confusing as it is a statement of the results before they are presented.

8. “Overall, our data show that the SIMULTANEOUS lack of all the forms....“

9. The labels in Figure 2F are not very clear. Perhaps using a different color (e.g. yellow) would allow better clarity.

10. On page 11, bottom: should be Fig. 3F

11.

Reviewer #3: In this study, the authors employ a zebrafish model to investigate the roles of Tfeb and Tfe3 in embryonic development and organogenesis through gene knockout. The results indicate that these transcription factors are crucial for early larval survival, and their loss leads to marked morphological abnormalities in organs such as the pancreas, liver, and intestine. Furthermore, the knockout embryos display heightened sensitivity to external stressors (such as oxidative stress and heat shock) underscoring the key role of Tfeb and Tfe3 in in vivo stress adaptation. Overall, the study addresses a significant topic; however, there remain several issues in data interpretation and experimental design that warrant further clarification and discussion.

Major points:

1. In Figure 1, the authors observe that female TKO zebrafish regain weight in subsequent stages, whereas males do not exhibit the similar pattern. Given this intriguing gender difference, the authors are encouraged to provide further discussion and experimental validation to elucidate the underlying molecular mechanisms.

2. In Figure 1E, only data on the length and weight of male zebrafish are presented, while the information of females was missing. To ensure data completeness and comparability, the authors should supplement or clarify the measurements for female samples.

3. The current data suggest that in the absence of Tfeb, Tfe3a plays a critical role in zebrafish embryogenesis. To clearly define the impact of Tfe3a deletion on embryonic development, the authors should incorporate additional experiments to determine whether its loss has a decisive effect. Have you examined the expression pattern of Tfeb, Tfe3a in multiple organs in zebrafish? Are they highly expressed in pancreas/live/gut than other organs, including brain, heart ?

4. In Figure 1C, the specific age of the zebrafish (ideally expressed in months) is not clearly indicated. The authors should provide precise developmental stage information to facilitate a better understanding of the temporal context of the results.

5. In Figure 1G, the wild-type group consists of fish aged 1.5 months, whereas the knockout group is 2 months old. Considering that age differences may influence the comparative results, the authors are advised to explain the rationale behind using different age groups or, if feasible, standardize the ages for both groups.

6. Figure 2D shows that Tfe3a transcripts (both long and short isoforms) are abundantly expressed in neural tissues and the gastrointestinal tract, yet the subsequent investigation primarily focuses on developmental defects in the pancreas and liver of the TKO zebrafish. The authors should further discuss this observation, addressing why the focus shifts to the pancreas and liver, and whether this expression pattern is linked to organ-specific functions.

In summary, while this study provides novel insights into the roles of Tfeb and Tfe3 in zebrafish development and stress responses, further clarification in experimental design and data interpretation is required. The authors are encouraged to address these issues comprehensively in their revision to enhance the scientific rigor and overall persuasiveness of the manuscript.

**Have all data underlying the figures and results presented in the manuscript been provided?**

Reviewer #1: Yes

Reviewer #2: Yes

Reviewer #3: Yes

PLOS authors have the option to publish the peer review history of their article (what does this mean? ). If published, this will include your full peer review and any attached files.

**Do you want your identity to be public for this peer review?** For information about this choice, including consent withdrawal, please see our Privacy Policy .

Reviewer #1: No

Reviewer #2: No

Reviewer #3: No

**Figure resubmission:**
---

## [Decision Letter · Decision Letter 1]

Dear Dr Puertollano,

We are pleased to inform you that your manuscript entitled "The transcription factors Tfeb and Tfe3 are required for survival and embryonic development of pancreas and liver in zebrafish" has been editorially accepted for publication in PLOS Genetics. Congratulations!

Yours sincerely,

Brian D. Perkins, Ph.D.

Academic Editor

PLOS Genetics

Pablo Wappner

Section Editor

PLOS Genetics

Aimée Dudley

Editor-in-Chief

PLOS Genetics

Anne Goriely

Editor-in-Chief

PLOS Genetics

Comments from the reviewers (if applicable):

Reviewer's Responses to Questions

**Comments to the Authors:**

Reviewer #1: The authors have addressed my concerns.

Reviewer #2: The authors have addressed all my concerns.

Reviewer #3: The authors have already addressed all my scientific questions. Now, this revised manuscript is suitable for the publication on Plos Genetics.

**Have all data underlying the figures and results presented in the manuscript been provided?**

Reviewer #1: Yes

Reviewer #2: Yes

Reviewer #3: Yes

PLOS authors have the option to publish the peer review history of their article (what does this mean? ). If published, this will include your full peer review and any attached files.

**Do you want your identity to be public for this peer review?** For information about this choice, including consent withdrawal, please see our Privacy Policy .

Reviewer #1: **Yes: ** Wen-Xing Ding

Reviewer #2: No

Reviewer #3: No

**Data Deposition**

http://datadryad.org/submit?journalID=pgenetics&manu=PGENETICS-D-25-00268R1

**Press Queries**

---

## [Editor Report · Acceptance letter]

PGENETICS-D-25-00268R1

The transcription factors Tfeb and Tfe3 are required for survival and embryonic development of pancreas and liver in zebrafish

Dear Dr Puertollano,

We are pleased to inform you that your manuscript entitled "The transcription factors Tfeb and Tfe3 are required for survival and embryonic development of pancreas and liver in zebrafish" has been formally accepted for publication in PLOS Genetics! Your manuscript is now with our production department and you will be notified of the publication date in due course.

With kind regards,

Olena Szabo

PLOS Genetics

On behalf of:
